# MEK inhibition reduced vascular tumor growth and coagulopathy in a mouse model with hyperactive GNAQ

Sandra Schrenk[1,2], Lindsay J. Bischoff [1,3], Jillian Goines[1], Yuqi Cai[1], Shruti Vemaraju[4,5], Yoshinobu Odaka [5,6], Samantha R. Good[1], Joseph S. Palumbo[2,7], Sara Szabo[8], Damien Reynaud [1,2], Catherine D. Van Raamsdonk [9], Richard A. Lang [4,5,10,11] & Elisa Boscolo [1,2] ✉

Activating non-inherited mutations in the guanine nucleotide-binding protein G(q) subunit alpha (GNAQ) gene family have been identified in childhood vascular tumors. Patients experience extensive disfigurement, chronic pain and severe complications including a potentially lethal coagulopathy termed Kasabach-Merritt phenomenon. Animal models for this class of vascular tumors do not exist. This has severely hindered the discovery of the molecular consequences of *GNAQ* mutations in the vasculature and, in turn, the pre-clinical development of effective targeted therapies. Here we report a mouse model expressing hyperactive mutant *GNAQ* in endothelial cells. Mutant mice develop vascular and coagulopathy phenotypes similar to those seen in patients. Mechanistically, by transcriptomic analysis we demonstrate increased mitogen activated protein kinase signaling in the mutant endothelial cells. Targeting of this pathway with Trametinib suppresses the tumor growth by reducing vascular cell proliferation and permeability. Trametinib also prevents the development of coagulopathy and improves mouse survival.

Vascular anomalies are defects caused by the abnormal development and/or growth of the vasculature. Hyperactive postzygotic (non-inherited) somatic mutations in the GNAQ gene (encoding the guanine nucleotide-binding protein G(q) subunit alpha or Gαq), or equivalent mutations in paralogous genes such as *GNA11* and *GNA14*, have been identified in a subset of vascular anomalies included in the classification proposed by the International Society for the Study of Vascular Anomalies (ISSVA)[1,2]. These include capillary malformation (CM)/Sturge-Weber syndrome (SWS), and a subset of vascular

tumors, including congenital hemangioma (CH), tufted angioma (TA), kaposiform hemangioendothelioma (KHE) and pyogenic granuloma (PG) (also known as lobular capillary hemangioma)[1–4]. With the exclusion of CM and SWS, which are mostly associated with *GNAQ* mutations affecting arginine 183 (p.R183), vascular tumors are linked to *GNAQ/11/14* mutation at glutamine 209 (p.Q209) (or equivalent p.Q205 in *GNA14*).

Congenital hemangiomas affect newborns and can persist throughout life (*e.g.* non-involuting congenital hemangioma or NICH).

[1]Division of Experimental Hematology and Cancer Biology, Cincinnati Children's Hospital Medical Center, Cincinnati, OH, USA. [2]Department of Pediatrics, University of Cincinnati College of Medicine, Cincinnati, OH, USA. [3]Department of Cancer Biology, University of Cincinnati College of Medicine, Cincinnati, OH, USA. [4]Science of Light Center, Cincinnati Children's Hospital Medical Center, Cincinnati, OH, USA. [5]The Visual Systems Group, Abrahamson Pediatric Eye Institute, Cincinnati Children's Hospital Medical Center, Cincinnati, OH, USA. [6]Department of Biology, University of Cincinnati Blue Ash College, Blue Ash, OH, USA. [7]Division of Hematology, Cincinnati Children's Hospital Medical Center, Cincinnati, OH, USA. [8]Department of Pathology and Laboratory Medicine, Cincinnati Children's Hospital Medical Center, Cincinnati, OH, USA. [9]Department of Medical Genetics, University of British Columbia, Vancouver, B.C., Canada. [10]Division of Developmental Biology, Cincinnati Children's Hospital Medical Center, Cincinnati, OH, USA. [11]Department of Ophthalmology, University of Cincinnati College of Medicine, Cincinnati, OH, USA. ✉e-mail: elisa.boscolo@cchmc.org

The other types of *GNAQ/11/14* p.Q209-related vascular anomalies make their appearance during childhood and subsequently undergo proliferative expansion in these patients. Despite different ages of onset and life cycle, these vascular lesions are characterized by overlapping histopathological features. These include abnormal, structurally irregular lobules of proliferative and tightly packed endothelial cells (EC), and mildly enlarged capillaries and/or venules. The complications of this class of vascular anomalies are infection, infiltration to adjacent tissues such as muscle and bone, and cardiac overload leading to high-output heart failure. In addition, patients with TA or KHE can develop severe consumptive coagulopathy and thrombocytopenia (Kasabach-Merritt phenomenon or KMP) that is potentially lethal. KMP was first described in 1940[5] in an infant affected by a rapidly enlarging vascular lesion associated with thrombocytopenia and hypofibrinogenemia, but the cellular and molecular events leading to KMP are still unknown. Current treatments for this subset of vascular anomalies include surgery, steroids, and vincristine[6], while recent clinical trials have investigated the use of Sirolimus[7,8]. While surgery is the most effective intervention, often it is not indicated because of the increased risk of bleeding in these patients. Furthermore, due to the proliferative nature of these vascular tumors, surgery may not be curative.

Gαq proteins (Gαq, Gα11, and Gα14) are members of the q class of G proteins that share about 90% sequence homology, and they mediate signals from G-protein-coupled receptors (GPCR) to phospholipase C-beta (PLC-β)[9]. They are normally expressed in different cell types, including vascular, blood and neuronal cells. Activated forms of $G\alpha_q$ and $G\alpha_{11}$ are frequent oncogenic drivers in uveal melanoma, an aggressive cancer of the adult eye, and they are also found in other melanocytic neoplasms[10]. As in vascular anomalies, these Gα subunits in uveal melanoma harbor a single amino acid substitution at Q209 or R183, which abrogates their intrinsic guanosine triphosphatase (GTPase) activity which normally serves to inactivate the protein. Therefore, gain-of-function, constitutively active mutants are thought to exist predominantly in the active, GTP-bound state. Studies in uveal melanoma have shown that this leads to hyperactivation of downstream effector molecules such as protein kinase C (PKC) and the mitogen-activated protein kinase (MAPK) cascade, which can lead to increased cellular proliferation[11]. To date, the EC-specific molecular and signaling consequences of hyperactive *GNAQ* are not well-defined, except for a handful of in vitro studies suggesting increased MAPK/ERK activity[2,4,12]. Furthermore, despite these genetic and mechanistic findings, genetic murine models for *GNAQ*-related vascular anomalies have not yet been reported.

Here, we set out to create a murine model to study the etiology and pathogenesis of gain-of-function *GNAQ*-related vascular anomalies. We investigated the vascular, hematological, and transcriptomic consequences of endothelial expression of *GNAQ* p.Q209L. We further assessed a link between vascular morphogenesis defects, proliferation, permeability, and increased MAPK/ERK activation, in the mouse model and in patient-derived tissue. Finally, to establish the importance of the MAPK/ERK signaling in vascular pathology, we performed proof-of-concept preclinical experiments with a MEK/ERK inhibitor to assess its efficacy in extending mouse survival and preventing vascular lesion formation, growth, and coagulopathy.

## Results

### Endothelial-specific *GNAQ*^Q209L^ expression during early postnatal development results in the formation of vascular abnormalities and vascular tufts

To study the effects of hyperactive mutant *GNAQ*^Q209L^ expression, we took advantage of a system that conditionally expresses human *GNAQ* p.Q209L in mice, the *Rosa26-floxed stop-GNAQ*^Q209L^ line[13]. We first examined the effects of constitutive *GNAQ*^Q209L^ expression by crossing *Rosa26-floxed stop-GNAQ*^Q209L^ mice to the ubiquitously expressed *CMV-Cre* transgenic mouse[14]. Embryonic lethality was observed for the *CMV-Cre; Rosa26-floxed stop-GNAQ*^Q209L^ genotype before embryonic day (E)8.5 (Supplementary Fig. 1, Supplementary Table 1).

Next, to test our hypothesis that endothelial *GNAQ* hyperactivation is sufficient to cause vascular anomalies, the *Rosa26-floxed stop-GNAQ*^Q209L^ mouse line was crossed with the vascular EC specific and tamoxifen-inducible *Cdh5-iCreER*^T2[15]^ (or *Pdgfb-iCreER*^T2[16]^, see Supplementary Figs. 4, 5, 7) to generate conditional *GNAQ*^Q209L^ expression in EC (hereafter called *iCdh5-GNAQ*^Q209L^). This system allows variable initiation of *GNAQ*^Q209L^ expression, which occurs when Cre expression is induced upon the first injection of tamoxifen. To investigate the early postnatal vascular phenotypes caused by *GNAQ*^Q209L^ in mice, we injected pups with tamoxifen at postnatal day 1 (P1) (Fig. 1a, b).

Human vascular anomalies caused by constitutively active *GNAQ* mutations are often localized to the skin or subcutaneous tissues[17–20]. Thereby, we harvested the subcutaneous tissue from the murine abdomen at P4, when 50% of mutant mouse lethality was detected (Fig. 1b). Macroscopically, we detected vascular abnormalities in the mutant *iCdh5-GNAQ*^Q209L^ mice (Fig. 1c). The whole-mount subcutaneous tissue of *iCdh5-GNAQ*^Q209L^ pups revealed abnormal and dilated CD31^+^ blood vessels (Fig. 1c and Supplementary Movies 1, 2). Vessel diameter and vascular area were increased in *iCdh5-GNAQ*^Q209L^ tissue compared to tamoxifen-injected littermates and mutant mice that did not receive tamoxifen (Fig. 1d). In mutant mice, blood vessels were disorganized and formed lobules of abnormal, structurally irregular, and tightly packed vascular lesions, hereafter referred to as "vascular tufts". For the analysis, we defined a vascular tuft as occupying an area ≥200 µm². Tuft number and area were significantly increased (both at $p < 0.0001$) in the *iCdh5-GNAQ*^Q209L^ tissues and were not detected in control tamoxifen-treated littermate tissue and in mutant mice that did not receive tamoxifen.

Some of the *GNAQ*-related vascular tumors such as pyogenic granuloma and kaposiform hemangioendothelioma are often located in the gastrointestinal tract[21–23]. Macroscopically we detected vascular lesions in the intestine (Fig. 1e). Therefore, we immunostained the murine intestinal *muscularis* for CD31. Whole-mount imaging analysis revealed the presence of vascular tufts in the intestinal *muscularis* of mutant mice (Fig. 1e). Vessel diameter, vascular area, tuft number and tuft area were significantly ($p = 0.0029$, $p = 0.0333$, $p = 0.0036$, and $p = 0.0181$ respectively) increased in *iCdh5-GNAQ*^Q209L^ mice when compared to the intestinal *muscularis* of control mice (Fig. 1f).

To test the efficiency and specificity of the *Cdh5-iCreER*^T2^ driver, we crossed this mouse line with the *Rosa26*^tdTomato^ lineage reporter (hereafter called *iCdh5-tdTomato*). Tamoxifen-induced activation of *Cdh5-iCreER*^T2^ led to 90.37 ± 7.9% expression of the recombined *Rosa26*^tdTomato^ reporter in the CD31^+^ vasculature of the subcutaneous tissue at P4 (Supplementary Fig. 2). Recombination outside of the CD31^+^ vasculature was not detected. Furthermore, we observed only minimal recombination (1.8 ± 0.88%) in *iCdh5-tdTomato* mice that did not receive tamoxifen (Supplementary Fig. 2).

To confirm that Gαq hyperactivation in EC induces vascular defects, we additionally employed a Gαq-DREADD mouse (DREADDs are designer receptors exclusively activated by designer drugs)[24]. The DREADD system consists of engineered G protein-coupled receptors (GPCR), which can precisely control GPCR signaling pathways such as Gαq, Gαs or Gαi[24]. Here, we used a Gαq DREADD mouse (*CAG-LSL-Gq-DREADD*) which is a system that expresses a modified M3 muscarinic receptor (hM3Dq)[25] which is only activated upon clozapine-N-oxide (CNO) administration. This mouse line was crossed with *Pdgfb-iCreER*^T2[16]^ (hereafter called *iPdgfb-hM3Dq*), to induce Gαq activity specifically in EC (Supplementary Fig. 3). Injection of tamoxifen on P1 and P2, followed by daily injection of CNO on P3-P8 caused an increase in CD31^+^ vessel density and vascular area in the

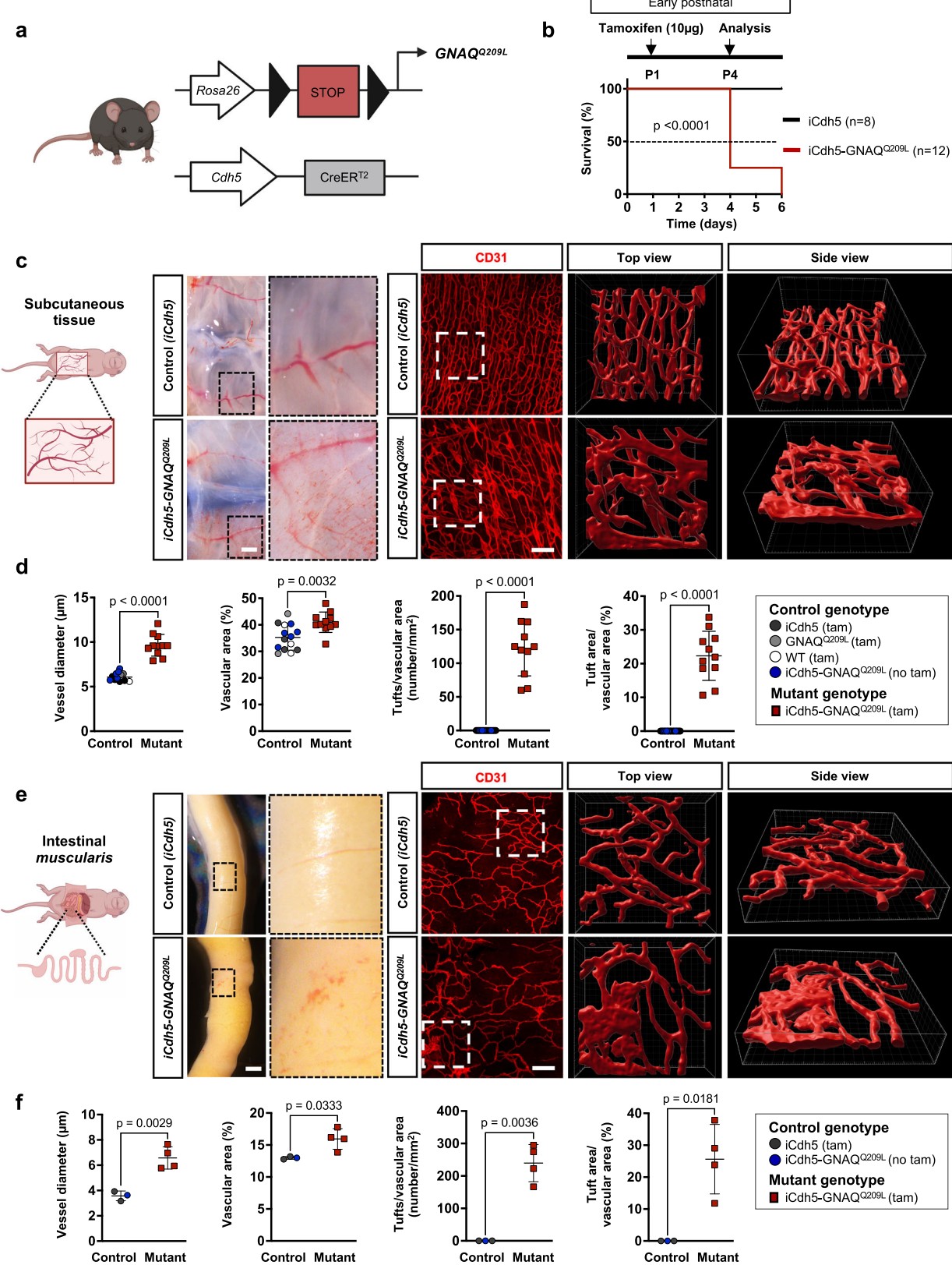

skin of *iPdgfb-hM3Dq* pups at P8 compared to littermate controls (Supplementary Fig. 3).

Additionally, we analyzed murine brain and retina vasculature. Tufts were detected in the vasculature of both the brain and retina of *iPdgfb-GNAQ^Q209L* mice (Supplementary Fig. 4, 5). Because *GNAQ*-related vascular anomalies primarily affect capillaries and veins[26–29], we

sought to determine which type of blood vessels were affected by vascular tuft formation. For this analysis, we utilized the developing vasculature of the retina as a model in which arteries and veins can be readily distinguished[30]. Analysis of mutant *iPdgfb-GNAQ^Q209L* mice at P8 showed that vascular tufts were exclusively localized to retinal veins and capillaries but absent in arteries (Supplementary Fig. 5).

**Fig. 1 | Early-onset vascular abnormalities and vascular tufts in *GNAQ^Q209L* expressing mice. a** Schematic describing breeding scheme for generating models of *GNAQ*-related vascular anomalies. The *Rosa26-floxed stop-GNAQ^Q209L* mouse was crossed with *Cdh5-iCreER^T2* to generate EC specific *GNAQ^Q209L* expression (mutant mice are thereafter called *iCdh5-GNAQ^Q209L*). **b** Schematic describing Tamoxifen (tam) regimen for postnatal (P1) induction before experimental analysis of vasculature in the subcutaneous and intestinal tissue at P4. Kaplan-Meier curve comparing the survival percentage of *iCdh5-GNAQ^Q209L* ($n = 12$, red) to *iCdh5* ($n = 8$, black) pups. Gehan–Breslow–Wilcoxon test ($p < 0.0001$). **c** Left panel: Representative low- and high-power photographs of subcutaneous tissue from tamoxifen-treated *iCdh5-GNAQ^Q209L* and control pup (*iCdh5*). Scale bar: 2 mm. Right panel: Representative confocal z-stack images (max. intensity projection) of whole-mount staining for CD31 (red). The boxed regions (white dashed lines) are enlarged, and 3D reconstructed. Scale bar: 100µm. **d** Quantification of vessel diameter, vascular area as percentage of total area, tuft number as well as tuft area normalized to vascular area. *iCdh5-GNAQ^Q209L* ($n = 11$) and controls ($n = 14$), mean±SD, unpaired two-tailed Welch's t-test. Mouse genotypes are indicated. **e** Left panel: Representative low- and high-power photographs of intestinal tissue from tamoxifen-treated *iCdh5-GNAQ^Q209L* and control pup (*iCdh5*). Scale bar: 2 mm. Right panel: Representative confocal z-stack images (max. intensity projection) of whole-mount intestinal *muscularis* stained for CD31 (red). The boxed regions (white dashed lines) are enlarged, and 3D reconstructed. Scale bar: 100µm. **f** Quantification of vessel diameter, vascular area as percentage of total area, tuft number, as well as tuft area normalized to vascular area. *iCdh5-GNAQ^Q209L* ($n = 4$) and controls ($n = 3$), mean±SD, unpaired two-tailed Welch's t-test. Mouse genotypes are indicated. Source data for (**b,d,f**) are provided as a Source Data file. Schematic in (**a,c,e**) created with Biorender.com.

## Endothelial *GNAQ^Q209L* expression in adult mice drives blood vessel dilation and hyperproliferation of EC

Here, we wanted to assess if expression of hyperactive *GNAQ* mutation in the adult vasculature can cause vascular defects. In these studies, we performed tamoxifen injections (two doses of 75 mg/kg) in mice at 6-8 weeks of age. Tissue was harvested one day before 50% of mutant mouse lethality was detected, which corresponded to day 6 after the start of tamoxifen (Fig. 2a). We did not detect a difference in survival between male and female *iCdh5-GNAQ^Q209L* mutant mice (Supplementary Fig. 6). *iCdh5-GNAQ^Q209L* mice developed vascular tufts and aberrant vascular morphogenesis in the subcutaneous tissues of the abdomen (Fig. 2b). Compared to control mice, the subcutaneous tissue of the *iCdh5-GNAQ^Q209L* mice showed dilated CD31^+ vessels, and significant (both $p < 0.0001$) increase of CD31^+ vascular density and vascular area (Fig. 2c). Furthermore, we analyzed vessel size distribution which revealed increased percentage of large (>100 µm²) vessels in mutant mice compared to controls (Fig. 2d).

Vascular tufts were also detected in the small intestine of *iCdh5-GNAQ^Q209L* mice at day 6 after tamoxifen induction (Fig. 2e). Whole-mount CD31 staining of the intestinal *muscularis* showed that the *iCdh5-GNAQ^Q209L* tissue had significantly ($p < 0.0001$) increased vessel diameter, vascular area, and tuft number/area, when compared to the intestinal *muscularis* of control mice (Fig. 2f, g).

Of note, we generated EC-specific expression of the mutant *GNAQ* p.Q209L in adult mice with the use of both *Cdh5-iCreER^T2* and *Pdgfb-iCreER^T2* and obtained similar phenotypes (Supplementary Fig. 7), demonstrating that the observed changes to the vasculature are not dependent on a specific endothelial Cre driver.

To test the efficiency and specificity of the *Cdh5-iCreER^T2* driver in adult mice, we analyzed *iCdh5-tdTomato* mice. Tamoxifen-induced activation of *Cdh5-iCreER^T2* in adult mice led to 88.13±2.93% and 91.62±1.12% expression of tdTomato in the CD31^+ vasculature in the subcutaneous and intestinal *muscularis* tissues, respectively (Supplementary Fig. 8). We did not detect recombination in non-endothelial cells (CD31^-). Furthermore, *iCdh5-tdTomato* mice that did not receive tamoxifen showed only minimal recombination in the subcutaneous and intestinal *muscularis* tissues (2.7 ± 2.50% and 2.20 ± 1.60%, respectively) (Supplementary Fig. 8).

Next, we sought to determine if vascular lesions are characterized by increased EC proliferation. To assess the proliferative capacity of EC expressing *GNAQ^Q209L*, we injected mice with EdU (5-Ethynyl-2′-deoxyuridine) to label cells undergoing DNA replication in S phase of the cell cycle. EdU was injected 5 days after tamoxifen induction and mouse tissue was analyzed 24 h after EdU administration (Fig. 2h). Quantification of EdU^+/CD31^+ cells in the intestinal *muscularis* or EdU^+/ERG^+ events in the subcutaneous tissue showed a significantly ($p = 0.0005$ and $p = 0.0043$, respectively) increased number of EdU^+ EC in *iCdh5-GNAQ^Q209L* mice compared to controls (Fig.2i, j; Supplementary Fig. 9 and Movies 3, 4).

## Increased vascular permeability in *iCdh5-GNAQ^Q209L* mice

Vascular lesions in patients, such as pyogenic granulomas, are prone to bleeding[31]. To investigate vascular permeability, we performed whole-mount staining of the intestinal *muscularis* with CD31 to label the vasculature and TER119 to label the erythrocytes (red blood cells, RBC) in adult mice, 6 days after tamoxifen administration (Fig. 3a, b). Quantification of extravasated RBC located outside of the CD31^+ vascular channels revealed the breakdown of vessel integrity in the *iCdh5-GNAQ^Q209L* mice, while only rare RBC were found outside of the vascular channels in control mice (Fig. 3b, c and Supplementary Movies 5, 6).

A previous study demonstrated that EC-specific deletion of *GNAQ/11* confers protection against VEGF-A-induced vascular permeability, implicating *GNAQ* in mediating this process[32]. We thereby hypothesized that hyperactive mutant *GNAQ* mice would demonstrate enhanced permeability compared to control mice in response to VEGF-A treatment. To analyze the VEGF-A-induced vascular permeability in our *iCdh5-GNAQ^Q209L* mice we performed a Miles assay[33] (Fig. 3d). We injected Evans Blue dye through the tail vein and assessed blue dye leakage from the vasculature in response to intradermal injection of PBS or VEGF-A. Quantification of the extravasated Evans blue dye revealed increased VEGF-A-induced vascular permeability in the skin of mutant mice compared to controls (Fig. 3e, f).

Endothelial permeability can be regulated by the stability of cell-cell adherens junctions composed by vascular endothelial (VE)-Cadherin[34]. Comparative assessment of VE-Cadherin expression in vivo was not possible because of the stark difference in blood vessel shape and lumen size between mutant and control mice. Thereby, we set out to generate an in vitro model of endothelial *GNAQ* p.Q209L by transducing endothelial colony-forming cells (ECFC)[35] with lentiviral constructs promoting doxycycline-inducible (i) expression of *GNAQ*-Q209L or *GNAQ*-WT (Fig. 3g). Immunoblotting revealed decreased expression of VE-Cadherin in the mutant iEC *GNAQ*-Q209L compared to control EC, iEC *GNAQ*-WT when gene expression was induced by treatment with doxycycline (Fig. 3h). Furthermore, in confluent cell monolayers, 48 h after doxycycline administration, VE-Cadherin expression at the cell junctions was visibly and significantly ($p = 0.0026$) reduced in the mutant EC compared to control *GNAQ*-WT EC (Fig. 3i, j).

## Mice expressing *GNAQ^Q209L* in the vasculature develop thrombocytopenia and severe coagulopathy, mimicking KMP in patients

Kasabach-Merritt phenomenon (KMP) is a poorly understood and life-threatening complication of vascular tumors that is characterized by thrombocytopenia and consumptive coagulopathy. KMP may affect up to 70% of all patients with KHE and 10-38% of patients with TA[19,36]. To study whether there is a similar complication in our *iCdh5-GNAQ^Q209L* mouse model, we collected blood and performed complete blood counts (CBC) on day 6 after tamoxifen injection (75 mg/kg) (Fig. 4a).

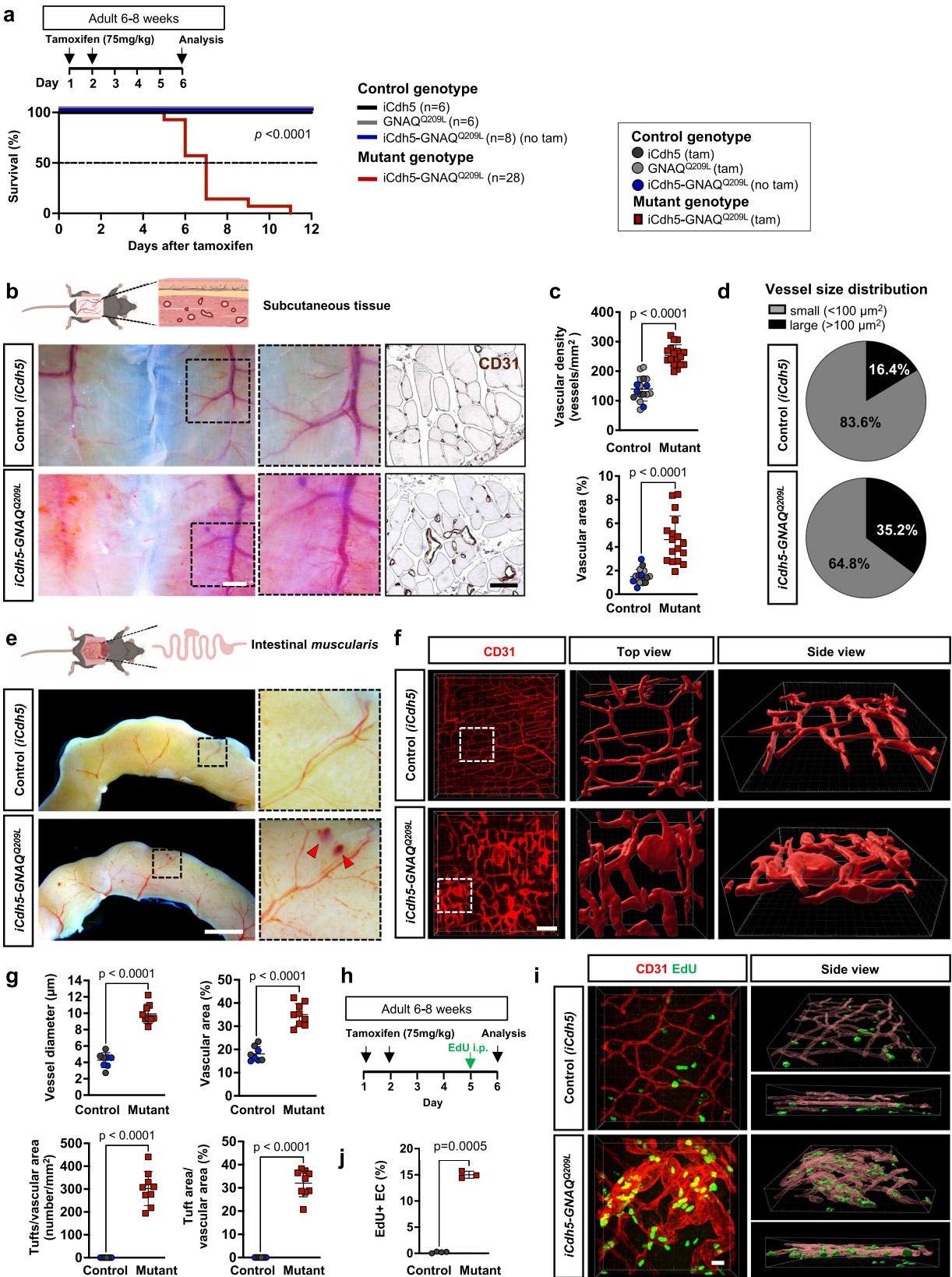

*iCdh5-GNAQ^Q209L* mice showed thrombocytopenia (low number of platelets), with a 42.2% reduction in platelet number compared to control animals. Mutant mice were also anemic, with 35.9% lower red blood cell (RBC) count and 34.5% lower hemoglobin levels, compared to tamoxifen-treated littermates and mutant mice that did not receive tamoxifen (Fig. 4b, c and Supplementary Fig. 10).

In patients, KMP presents with elevated D-Dimer levels[36], a marker of fibrin deposition and subsequent fibrinolysis. *iCdh5-GNAQ^Q209L* plasma analysis revealed an average 3.6-fold increase in D-Dimer levels compared with control animals (Fig. 4d). Prothrombin time (PT) and activated partial thromboplastin time (aPTT) analysis showed that PT was significantly reduced ($p = 0.0006$) in mutant

**Fig. 2 | EC-specific *GNAQ*[Q209L] expression in adult mice results in increased blood vessel density, abnormal morphogenesis, and EC hyperproliferation.**
**a** Schematic describing Tamoxifen regimen for adult mice (6-8 weeks) before experimental analysis of vasculature in subcutaneous and intestinal tissue at day 6 after tamoxifen induction. Kaplan-Meier curve comparing survival percentage of tamoxifen-induced *iCdh5-GNAQ*[Q209L] (n = 28, red) to controls (n = 20, black, grey, blue). Mouse genotypes are indicated. Gehan−Breslow−Wilcoxon test (p < 0.0001). **b** Left panel: Representative low- and high-power photographs of the subcutaneous tissue of tamoxifen-treated *iCdh5-GNAQ*[Q209L] and *iCdh5* control mouse. Scale bar: 2 mm. Right panel: Representative images of subcutaneous tissue sections immuno-stained for CD31 (dark brown), scale bar: 50μm. **c** Quantification of vascular density defined as the number of vessels per area and vascular area as percentage of total area. *iCdh5-GNAQ*[Q209L] (n = 17) and control animals (n = 17). **d** Vessel size distribution in subcutaneous tissue sections. Percentage of small (<100 μm²) and large (>100 μm²) vessels. **e** Representative low- and high-power photographs of intestine of tamoxifen-treated *iCdh5-GNAQ*[Q209L] and *iCdh5* control mouse. Vascular lesions are indicated by red arrowheads. Scale bar: 2 mm. **f** Representative confocal z-stack images (max. intensity projection) of whole-mount intestinal *muscularis* stained for CD31 (red). The boxed regions (white dashed lines) are enlarged, and 3D reconstructed. Scale bar: 100 μm. **g** The intestinal *muscularis* vasculature was quantified for vessel diameter, vascular area as percentage of total area, tuft number, as well as tuft area normalized to vascular area. *iCdh5-GNAQ*[Q209L] (n = 9) and control littermates (n = 8). **h** Tamoxifen induction and EdU injection scheme. EdU was administered to *iCdh5-GNAQ*[Q209L] and control littermates, 24 h before analysis. **i** Representative confocal z-stack images (max. intensity projection) of intestinal *muscularis* whole-mount preparations labeled for EdU (green) and CD31 (red). Images were 3D reconstructed. Scale bar: 20μm. **j** The number of EdU positive endothelial cells (EdU⁺/CD31⁺) in *iCdh5-GNAQ*[Q209L] (n = 3) and control littermates (n = 4) was counted and shown as percentage of total number of endothelial cells (CD31⁺). Data in (**c,g,j**) represented as mean±SD, unpaired two-tailed Welch's t-test. Mouse genotypes are indicated. Source data for (**a,c,d,g,j**) provided as a Source Data file. Schematic in (**b,e**) created with Biorender.com.

---

mice (Supplementary Fig. 11a). We further analyzed blood smears and quantified polychromasia and schistocytes. Polychromasia was significantly increased (p = 0.0178) in mutant mice compared to tamoxifen-treated control littermates and mutant mice without tamoxifen, while schistocytes were low in all groups (Supplementary Fig. 11b, c). These data suggest that the anemia was not driven by intravascular hemolysis or red blood cell aplasia.

One proposed mechanism for the association of vascular anomalies with systemic thrombocytopenia is the accumulation and sequestration of platelets and fibrin/fibrinogen in the malformed vessels[37]. To assess for platelet accumulation in the vascular tufts, we perfused the mouse prior to tissue collection and stained the intestinal *muscularis* for CD41 or CD42b, which are markers for platelets. CD41 and CD42b immunostaining was highly enriched in the CD31⁺ vascular tufts in the mutant *iCdh5-GNAQ*[Q209L] mice even upon tissue perfusion, indicating increased adherence to the vessel wall, while being almost absent from perfused control tissue (Fig. 4e, f and Supplementary Movies 7–10). In the subcutaneous tissue we obtained a similar pattern of platelet localization (Supplementary Fig. 12a–c). Additionally, we also detected increased fibrin/fibrinogen deposition in the vascular tufts of *iCdh5-GNAQ*[Q209L] mice, compared to control mice (Supplementary Fig. 12d).

To determine if hematopoietic alterations also contribute to the systemic thrombocytopenia and anemia, we performed quantitative analysis of the murine hematopoietic stem and progenitor compartments with a 15-fluorochrome flow cytometry protocol[38,39]. We did not observe differences in the bone marrow cellularity, the absolute number of hematopoietic stem cell or early multipotent progenitors between the *iCdh5-GNAQ*[Q209L] mutant mice and the control mice (Supplementary Fig. 13). We detected some subtle variations in the number of late progenitors for the megakaryocytic and erythroid lineages which are most likely associated with compensatory mechanisms to rescue the peripheral thrombocytopenia and anemia.

### *GNAQ* p.Q209L expression in endothelial cells drives transcriptional activation of angiogenesis, MAPK signaling, inflammatory response, and coagulation

While some effectors of the hyperactive mutant *GNAQ* p.Q209L have been discovered in the context of uveal melanoma[10,40–45], it is still unclear which EC-specific *GNAQ* effectors are driving the formation and expansion of vascular abnormalities.

To identify the pathways that are modulated by *GNAQ* p.Q209L expression in EC, we performed transcriptional profiling of human ECFC expressing wild-type (WT) and mutant *GNAQ* (n = 4 biological replicates for cell type). The principal component analysis revealed a clear separation between iEC *GNAQ*-Q209L and iEC *GNAQ*-WT, indicating significant transcriptional alterations. Importantly, iEC

*GNAQ*-WT with and without doxycycline were similar, showing that *GNAQ*-WT overexpression has little effect on the transcriptional changes observed in the mutant cells (Fig. 5a). We found that a total of 917 genes were differentially expressed in iEC *GNAQ*-Q209L compared to iEC *GNAQ*-WT, with more up-regulated genes than down-regulated genes (Fig. 5b, c). KEGG (Kyoto Encyclopedia of Genes and Genomes) and GO-BP (Gene Ontology Biological Pathways) pathway analysis of the differentially expressed genes identified significant changes in multiple pathways (log2 fold change of ≥1.0 or ≤−1.0 and adjusted q-value ≤0.05). Among these, we identified angiogenesis, MAPK signaling, inflammatory response and complement and coagulation pathways (Fig. 5d). These data were confirmed with Gene Set Enrichment Analysis (GSEA) for the Hallmarks: KRAS signaling up, Angiogenesis, Inflammatory response, TNFα signaling via NFkB, and Complement (Supplementary Fig. 14). Furthermore, direct analysis of mRNA expression by qPCR confirmed the modulation of genes in these pathways (Fig. 5e).

To validate the upregulation of MAPK signaling and the angiogenic growth factor angiopoietin-2 (ANGPT2) expression at the protein level in the mutant EC expressing *GNAQ*-Q209L, we performed immunoblotting (Fig. 5f). Doxycycline administration induced protein expression of GNAQ in a dose-dependent manner in both *GNAQ*-WT and *GNAQ*-Q209L expressing cells. Importantly, only mutant iEC *GNAQ*-Q209L showed increasing and doxycycline dose-dependent levels of activated (phosphorylated) ERK, confirming the MAPK signaling pathway activation, while phosphorylated AKT was not upregulated in the GNAQ mutant EC. ANGPT2 protein expression levels were also increased. Conversely, no ERK activation or ANGPT2 expression were noted in the iEC *GNAQ*-WT in response to doxycycline administration (Fig. 5f).

### Increased endothelial MAPK/ERK signaling in murine and patient-derived tissue expressing *GNAQ*[Q209L]

To determine that expression of *GNAQ*[Q209L] in the vasculature in vivo promotes activation of the MAPK/ERK pathway and increases proliferation we analyzed the subcutaneous tissue of adult mice.

The subcutaneous tissue sections were immunostained for ERG to label the EC nuclei, for phospho(p)-ERK and for Ki67 to label cells with increased ERK activation and proliferative capacity, respectively (Fig. 6a, b). The number of EC per field area was expressed as the number of ERG + cells/mm² and was significantly (p = 0.0002) increased in *iCdh5-GNAQ*[Q209L] tissue compared to control (Fig. 6c). The percentage of pERK⁺ ECs and Ki67⁺ ECs were also significantly (p = 0.0285 and p = 0.0185) increased. There was also a trend of increased number of double-positive pERK⁺/Ki67⁺ ECs in the *iCdh5-GNAQ*[Q209L] tissues compared to controls (Fig. 6c).

To confirm the relevance of these results, we also analyzed a patient-derived cutaneous vascular tumor with a somatic *GNAQ*

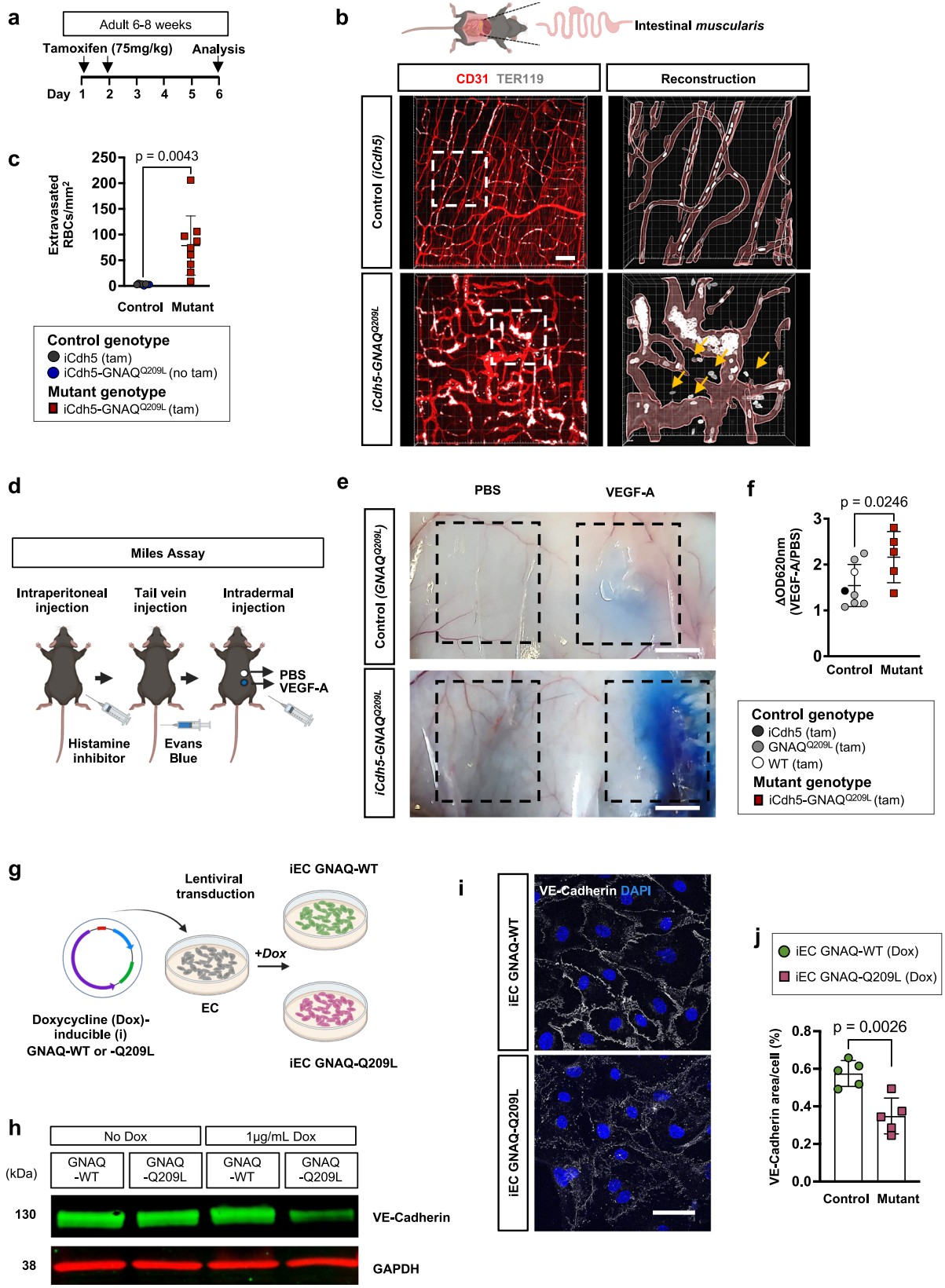

p.Q209L mutation (Fig. 6d). The tumor tissue showed numerous blood vessels and tufts of EC as shown by staining with the human-specific endothelial lectin Ulex Europaeus Agglutinin-I (UEAI) (Fig. 6e). The patient tissue was compared to control human neonatal foreskin from 5 different donors. Quantification of the UEAI⁺ vascular area showed that it was increased in the lesional patient tissue compared to normal foreskin (Fig. 6f, g). Furthermore, the number of pERK⁺ ECs, Ki67⁺ ECs and double-positive (pERK⁺/Ki67⁺) ECs were higher in the patient tissue compared to control human foreskin from 5 different donors (Fig. 6g).

**Fig. 3 | EC-specific *GNAQ^{Q209L}* expression results in vascular hyperpermeability and decreased VE-Cadherin expression. a** Tamoxifen induction scheme. **b** Representative confocal z-stack images of intestinal *muscularis* whole-mount preparations stained for erythrocytes (TER119 positive cells−white) and CD31 positive endothelium (red). The boxed regions (white dashed lines) are enlarged, 3D reconstructed, and show extravasated TER119^+ erythrocytes (yellow arrows). Scale bar: 70 µm. **c** Quantification of the number of extravasated TER119^+ erythrocytes normalized to tissue area. *iCdh5-GNAQ^{Q209L}* (*n* = 9) and control littermates (*n* = 8), mean±SD, unpaired two-tailed Welch's t-test. Mouse genotypes are indicated. **d** Diagram of experimental procedure for Miles assay. Mice were pretreated with histamine inhibitor followed by intravenous injection of Evans blue dye. This was followed by an intradermal injection of PBS with or without VEGF (100 ng). **e** Representative photographs of the leakage of Evans Blue dye in dorsal flank skin of *iCdh5-GNAQ^{Q209L}* and control littermate (*GNAQ^{Q209L}*) that were subjected to Miles assay. Scale bar: 500µm. **f** Quantification of vascular leakage in the Miles assay. The absorbance (optical density, OD620nm) from each skin sample was normalized to the weight of the tissue sample. *iCdh5-GNAQ^{Q209L}* (*n* = 5) and control littermates (*n* = 8), mean±SD, unpaired two-tailed Welch's t-test. Mouse genotypes are indicated. **g** Schematic of lentiviral transduction of human endothelial cells. The expression of *GNAQ*-Q209L or *GNAQ*-WT was induced by doxycycline (dox) treatment (1 µg/mL). **h** Immunoblot for VE-Cadherin expression in iEC *GNAQ*-Q209L or *GNAQ*-WT after doxycycline treatment for 48 h. Representative of 2 independent experiments. **i** Representative confocal z-stack images (max. intensity projection) of confluent cell monolayer stained for VE-Cadherin (white). Nuclei were labeled with DAPI (blue). Expression was compared between iEC *GNAQ*-WT and iEC *GNAQ*-Q209L upon 1 µg/mL dox treatment. Scale bar: 50µm. **j** VE-Cadherin positive area was quantified and is shown as percentage of positive area per cell, *n* = 5 independent experiments per group, mean±SD, unpaired two-tailed Welch's t-test. Source data for (**c,f,h,j**) are provided as a Source Data file. Schematic in (**b,d,g**) created with Biorender.com.

## Trametinib treatment rescues vascular phenotype, EC hyperproliferation, thrombocytopenia, and prolongs survival of *GNAQ* mutant mice

To investigate the role of Gαq downstream effector MAPK/ERK in the pathogenesis of vascular anomalies in our *GNAQ^{Q209L}* mouse model, we performed proof-of-concept experiments with the MEK/ERK inhibitor, Trametinib (MEKINIST®). First, we tested the efficacy of Trametinib in a preventative scheme. We performed tamoxifen injections (2 daily doses of 75 mg/kg) in 6-week-old *iCdh5-GNAQ^{Q209L}* mutant mice, while also delivering Trametinib (2 mg/kg) or vehicle, once daily (Fig. 7a). On day 5 (corresponding to 90% survival rate of mutant *iCdh5-GNAQ^{Q209L}* mice), tissues were harvested. To evaluate the ability of Trametinib to antagonize the development of vascular anomalies, subcutaneous tissue and intestinal *muscularis* were analyzed. Macroscopic images and CD31 immunostaining revealed decreased vascularity in the subcutaneous tissue sections of Trametinib-treated mutant mice compared to vehicle-treated mice. (Fig. 7b). Trametinib-treated mutant mice showed reduced percentage of large vessels (15.4 ± 3.4%) compared to vehicle-treated (25.7 ± 7.3%) mutant mice (Fig. 7c) and reduced vascular density and vascular area, which reached values similar to unchallenged control mice (see dotted lines) (Fig. 7d).

Intestinal *muscularis* was analyzed macroscopically and by whole-mount immunofluorescence CD31 staining (Fig. 7e). Trametinib treatment normalized CD31^+ vessel diameter and vascular area compared to vehicle-treated *iCdh5-GNAQ^{Q209L}* mice (Fig. 7f). Furthermore, the Trametinib-treated mice showed a significant (*p* = 0.0082 and *p* < 0.0001) reduction in the number of tufts, as well as tuft area (Fig. 7f).

To determine if Trametinib affects the proliferative capacity of EC in the mutant mice, we injected mice with EdU 24 h before the analysis at day 5 (see schematic in Fig. 7a). Quantification for EdU^+/CD31^+ events showed a significant (*p* < 0.0001) decrease in the number of EdU^+ EC in the intestinal *muscularis* (Fig. 7g, h and Supplementary Movies 11, 12) and in the subcutaneous tissue of Trametinib-treated *iCdh5-GNAQ^{Q209L}* mice compared to vehicle-treated (Supplementary Fig. 15).

Trametinib treatment additionally rescued the vascular permeability of *iCdh5-GNAQ^{Q209L}* mice (Fig. 7i). The number of extravasated TER119^+ erythrocytes was significantly reduced (*p* = 0.0036) in Trametinib-treated mice compared to vehicle-treated and reached values similar to unchallenged control mice (Fig. 7j).

Furthermore, blood cell analysis revealed a rescue of the platelet number in the Trametinib-treated *iCdh5-GNAQ^{Q209L}* mice (Fig. 7k, Supplementary Fig. 16), reaching comparable levels to unchallenged control mice (see dotted line) and significantly (*p* = 0.0386) higher than vehicle-treated mice. Plasma analysis showed that D-Dimer levels were significantly (*p* = 0.0086) lower in Trametinib-treated *iCdh5-GNAQ^{Q209L}* mice, thereby Trametinib prevented the onset of coagulopathy (Fig. 7k).

The efficacy of Trametinib in inhibiting the MAPK/ERK pathway in iEC *GNAQ*-Q209L was confirmed by immunoblotting. Treatment of iEC GNAQ-Q209L with Trametinib (1 and 5 nM) for 24 h suppressed ERK activation in a dose-dependent manner (Fig. 7l). Next, to identify the MEK-dependent targets of hyperactive mutant *GNAQ*, we performed transcriptional profiling in iEC *GNAQ*-Q209L treated with Trametinib or vehicle and compared them to vehicle-treated iEC *GNAQ*-WT control cells (Fig. 7m). We identified 617 genes whose expression was upregulated in iEC *GNAQ*-Q209L compared to iEC *GNAQ*-WT. The expression levels of 73 of these genes were restored to normal levels in response to Trametinib treatment. Among these we identified genes implicated in the MAPK pathway activity, inflammatory response and Notch signaling (Fig. 7m). GSEA revealed that Trametinib treatment resulted in the normalization of upregulated genes associated with hallmark gene signatures of KRAS Signaling Up, Angiogenesis, TNFα signaling via NFkB, Inflammatory response, Complement, and Coagulation (Supplementary Fig. 17).

Lastly, we performed pre-clinical treatment studies to determine the therapeutic efficacy of Trametinib in promoting the survival of mutant mice. For these studies we used a lower dose of tamoxifen (40 mg/kg) which resulted in 50% lethality at day 13-15 (Supplementary Fig. 18). Daily Trametinib treatment started 8 days after tamoxifen induction, and significantly (*p* = 0.0014) extended the life span of the mutant mice of up to 9 days compared to the vehicle-treated mice (Fig. 7n). Vehicle or Trametinib treatment of genetic controls did not result in illness or morbidity.

## Discussion

Somatic mutations in *GNAQ/11/14* that cause gain-of-function constitutively active Gαq signaling have been identified in a variety of vascular anomalies, including malformations and tumors. Despite these findings, genetic animal models for Gαq driven vascular anomalies have not yet been reported. Therefore, the cellular and molecular determinants of constitutively active *GNAQ* signaling in blood vessel dysmorphogenesis have not been investigated. In this study, we generated a murine model of mutant *GNAQ*-driven vascular tumors by conditionally expressing *GNAQ^{Q209L}* in *Cdh5* (or *Pdgfb*) expressing ECs. This model recapitulated common histopathological findings in vascular tumors with *GNAQ/11/14* mutations. These include the formation of vascular tufts which are lobules of proliferative and irregularly structured vascular lesions. Vascular tufts in mutant mice were characterized by hyperproliferation and increased permeability. Furthermore, our *GNAQ^{Q209L}* mouse model developed a coagulopathy that resembles KMP, a life-threatening coagulopathy seen in some of these patients. Lastly, we investigated the transcriptomic effects of *GNAQ^{Q209L}* signaling and determined that angiogenesis, MAPK/ERK signaling, coagulation and inflammation pathways are increased. In proof-of-concept studies, we showed that Trametinib, a MEK/ERK inhibitor, rescues the aberrant vascular morphogenesis, hyperproliferation, permeability, KMP-like coagulopathy, and extended mutant

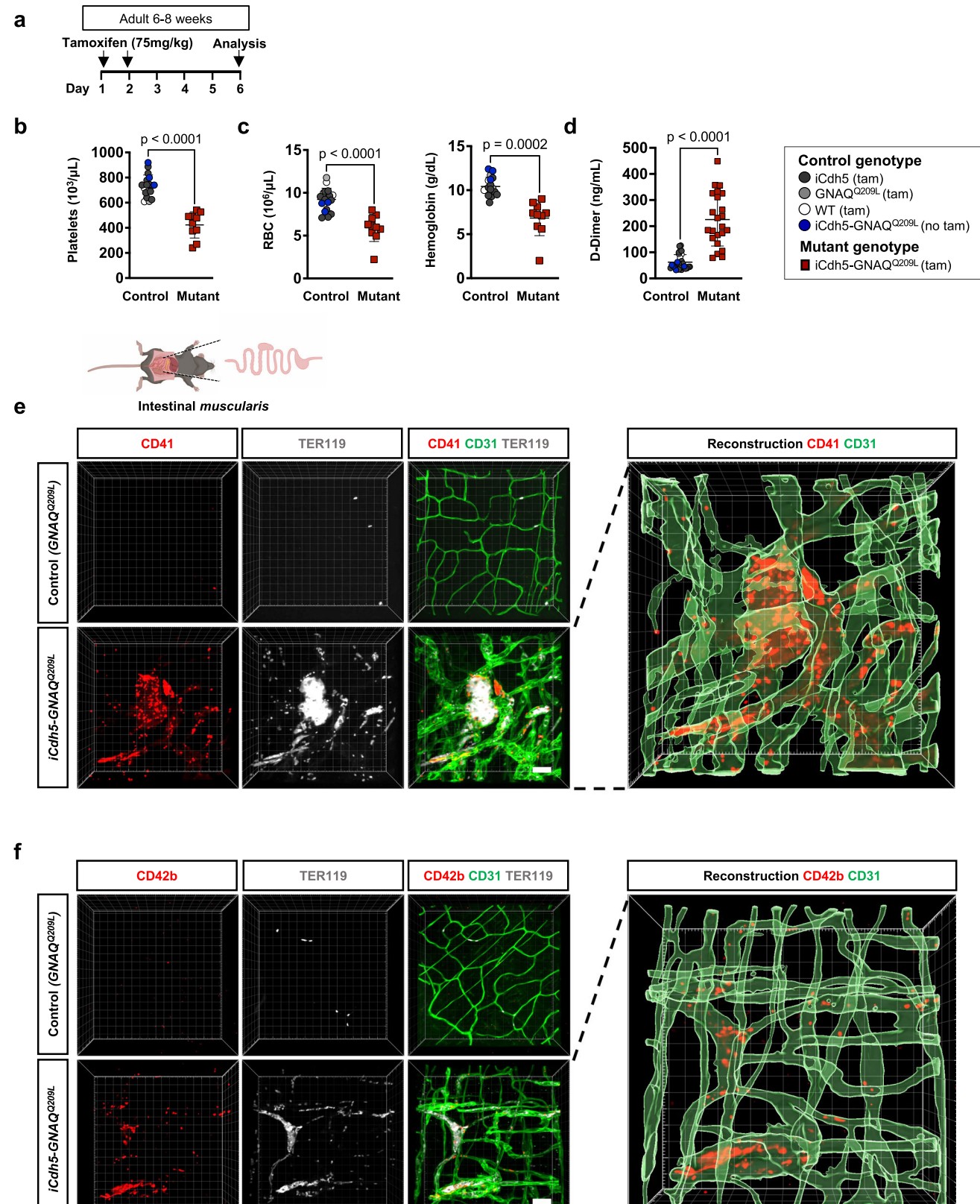

**Fig. 4 | Mice expressing *GNAQ^Q209L* in the vasculature develop thrombocytopenia and severe coagulopathy. a** Tamoxifen induction scheme. **b-c** Hematological parameters were measured in *iCdh5-GNAQ^Q209L* (*n* = 10) and controls (*n* = 18), mean±SD, unpaired two-tailed Welch's t-test. Mouse genotypes are indicated. **b** Platelet counts, **c** RBC (red blood cell counts), and hemoglobin levels. **d** D-Dimer levels were measured in plasma samples from *iCdh5-GNAQ^Q209L* (*n* = 23) and controls (*n* = 28), mean±SD, unpaired two-tailed Welch's t-test. Mouse genotypes are indicated. **e-f** Representative confocal z-stack images (max. intensity projection) of intestinal *muscularis* stained for **e** CD41 or **f** CD42b (red) together with CD31 (green) and TER119 (white/grey). Images are representative of *n* = 3 mice/group and were 3D reconstructed. Scale bar: 40μm. Source data for (**b,c,d**) are provided as a Source Data file. Schematic in (**e**) created with Biorender.com.

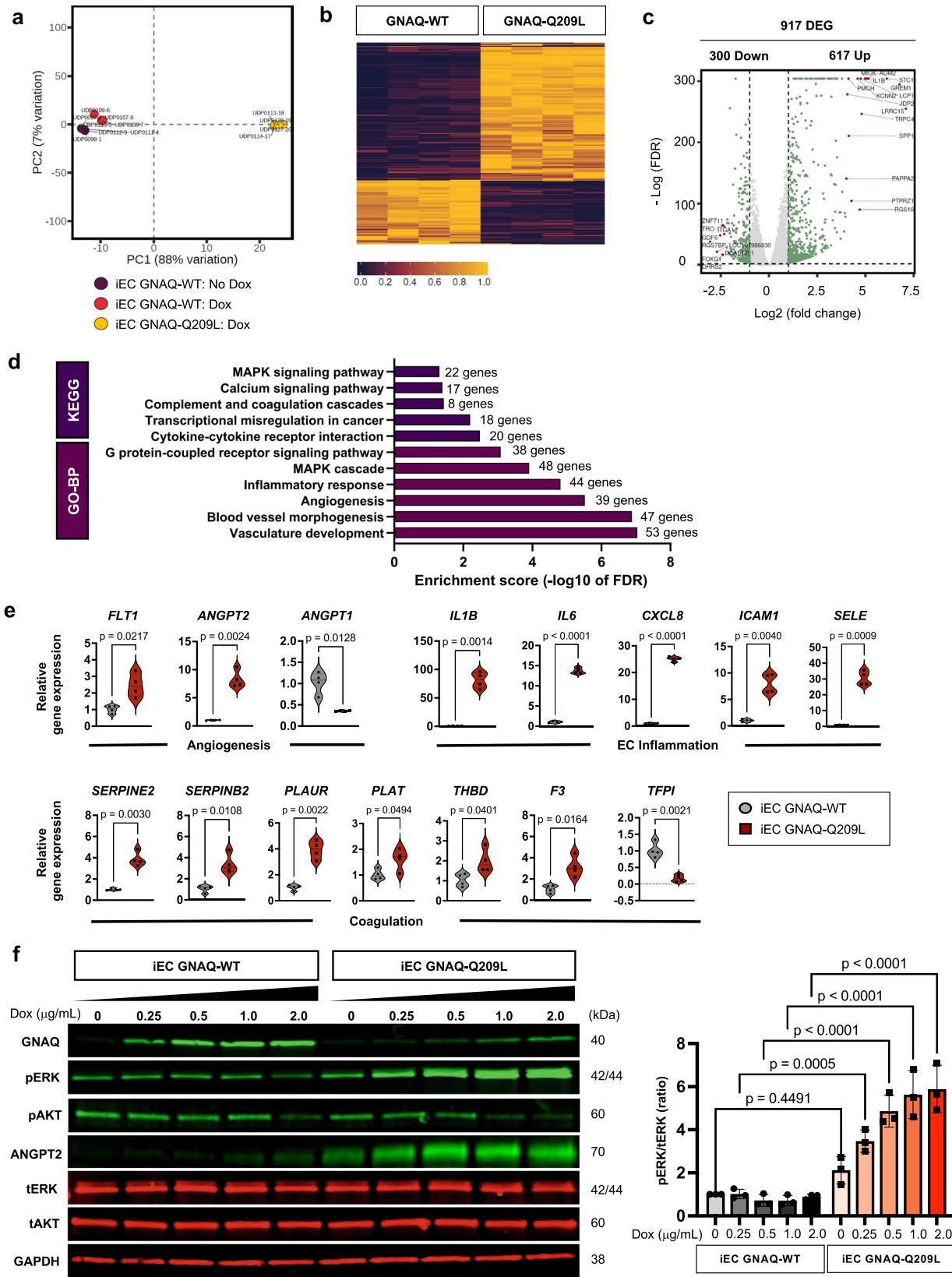

mouse survival, suggesting that MAPK/ERK signaling drives these phenotypic manifestations.

Our murine model was generated with the use of inducible EC-specific Cre-drivers to study the effects of constitutively active *GNAQ* in the endothelium. In patients, *GNAQ/11/14* mutations are somatic (*i.e.*, non-inherited) and the allelic frequency in the patients' affected tissue is generally quite low. Studies in congenital hemangioma and capillary malformation reported enrichment of the *GNAQ* mutation in the EC population, strongly suggesting the mutations originate in EC[1,46,47]. To drive expression of *GNAQ^{Q209L}* in EC we used *Cdh5-iCreER^{T2}* or *Pdgfb-iCreER^{T2}* *Cre* driver murine lines. It is worth noting that we did not detect major phenotypical differences between

**Fig. 5 | RNA-Sequencing of iEC *GNAQ*-Q209L reveals enriched expression of genes related to angiogenesis, MAPK signaling, inflammatory response, and coagulation. a** Principal Component Analysis (PCA) of RNA-sequencing data obtained from iEC *GNAQ*-WT no Dox, iEC *GNAQ*-WT + Dox and iEC *GNAQ*-Q209L + Dox. **b** iEC *GNAQ*-WT and to *GNAQ*-Q209L (n = 4 biological replicates per group) were treated with 1 μg/mL doxycycline for 24 h and total RNA was collected for RNA-seq analysis. Differentially expressed genes (DEG) are defined as log2 (fold change) ≥1 and adjusted *P* value of <0.05. Heat maps of differentially expressed transcripts in iEC *GNAQ*-Q209L compared to *GNAQ*-WT. Wald test for hypothesis testing. **c** Volcano-plot of DEGs in iEC *GNAQ*-Q209L compared to *GNAQ*-WT. A total of 617 genes showed increased expression while 300 genes showed decreased expression in iEC *GNAQ*-Q209L compared to iEC *GNAQ*-WT. **d** KEGG and GO-BP pathway analyses of DEGs. **e** A selected group of DEGs were validated using RT-qPCR. n = 4 samples/group *(*median and quartiles), unpaired two-tailed Welch's t-test. **f** Representative immunoblots of GNAQ, phospho-ERK (pERK, Thr202/Tyr204), total ERK (tERK), phospho-AKT (pAKT, Ser473), total AKT (tAKT), ANGPT2, and GAPDH in iEC *GNAQ*-WT and *GNAQ*-Q209L 48 h after doxycycline induction (0, 0.25, 0.5, 1 and 2 μg/ml of Dox). The expression of pERK was quantified and normalized to tERK expression (*n* = 3 independent experiments). Mean ±SD, one-way ANOVA with Tukey's multiple comparisons. P-values for statistical analysis between all groups are listed in the Source data file. Source data for (**c,d,e,f**) are provided as a Source Data file.

*iCdh5-GNAQ*$^{Q209L}$ and *iPdgfb-GNAQ*$^{Q209L}$ animals. The presence of somatic activating *GNAQ* mutations in vascular anomalies suggests an essential role for Gαq signaling in vascular development and homeostasis. Here, we show that the *GNAQ* mutation p.Q209L is sufficient for the formation of hyperproliferative vascular lesions when driven solely in EC.

Previous studies have shown that deficiency of Gαq and/or Gα11 in endothelial cells resulted in reduced EC proliferation and impaired retinal angiogenesis, while conferring protection against VEGF-A-induced vascular permeability[32]. In our study we further demonstrate the essential role of *GNAQ* in vascular permeability and show that gain-of-function *GNAQ* mutations can promote vascular leakage and disrupt VE-Cadherin expression in adherens junctions. Taken together, this demonstrates that both gain-of-function and loss-of-function *GNAQ* mutations can result in abnormal angiogenesis.

*GNAQ/11* gain-of-function mutations are very frequent in uveal melanoma, an aggressive tumor of melanocytes in the uveal tract of the eye. In this context, it has been found that *GNAQ*$^{Q209L}$ can lead to hyperactivation of the downstream MAPK/ERK signaling pathway as well as activation of the Hippo pathway through nuclear localization of YAP1 via a Trio-Rho/Rac signaling circuit[11]. To date, MAPK inhibitors have not shown clinical benefit for uveal melanoma patients, making the role of MAPK in uveal melanoma less clear. In fact, many studies rely on poorly characterized melanoma cell lines that express multiple mutation types[48,49], while studies with primary uveal melanomas reported very heterogeneous levels of ERK activation, suggesting there is no association with *GNAQ/11* mutations[50,51]. In our studies, we generated transcriptomic data to show that MAPK and KRAS pathways are among the most upregulated signaling pathways in EC expressing *GNAQ*$^{Q209L}$. We also went on to confirm increased pERK levels in tissues of our *iCdh5-GNAQ*$^{Q209L}$ animal model and in a patient-derived vascular tumor tissue with a confirmed somatic *GNAQ* p.Q209L mutation. Although different *GNAQ* mutations are associated with different classes of vascular anomalies (such as *GNAQ* R183Q and *GNAQ* Q209R in CM and *GNAQ* Q209L in vascular tumors), recent studies highlighted that the transcriptional consequences of these mutations in ECs are similar and include upregulation of pathways such as MAPK, angiogenesis, inflammation via TNFα and NFkB, and upregulation of *ANGPT2*[12,52]. Published data also suggest that the differences between these *GNAQ* mutation types affect the level of activation/expression rather than the specific downstream targets[12]. Furthermore, similar to a recent study in melanocytes, we did not detect significant upregulation of genes associated with the non-canonical activation of the Hippo pathway[42].

To date, only few direct inhibitors of Gαq/11 have been reported. Among these are YM-254890 and FR900359, which are natural products isolated from the bacterium *Chromobacterium* and from the plant *Ardisia crenata Sims*. While these compounds showed efficacy in inhibiting abnormal signaling in Gαq mutant uveal melanoma cell lines, their complex chemical structures have hindered their commercial development[41,53,54]. For this reason, additional pathways and effectors have been investigated and shown to be important for uveal melanoma tumorigenesis such as PKC, Trio/Rho/Rac, YAP/TAZ, and FAK[44,55,56]. Several studies have shown the efficacy of mono- or -dual therapy in laboratory models. However, targeted therapy in patients is still an unmet clinical need. While the role of these effectors has not yet been explored in *GNAQ*-related vascular anomalies, our results show that MEK/ERK inhibition with Trametinib in the murine model prevented vascular lesions and the associated complications such as coagulopathy, suggesting this pathway is implicated in the disease pathogenesis. Recently published reports have shown that the hyperactive RAS/MAPK signaling is implicated in other vascular tumors and malformations[57] and some authors successfully modeled mouse vascular anomalies associated with KRAS-G12D/V activating mutations[58–60]. These models showed improvement of the vascular phenotype upon treatment with Trametinib. Clearly, RAS/MAPK signaling is implicated in the manifestation of vascular anomalies, and as such it will be a critical area for future drug development. However, it is important to notice that in our study, while Trametinib treatment significantly improved survival compared to vehicle treatment, mice failed to thrive beyond day 18. In vascular anomalies, this could be explained by the activity of MAPK-independent pathways downstream of the mutant *GNAQ*. Among the 617 genes upregulated in the EC expressing *GNAQ*-Q209L, Trametinib treatment normalized the expression of 73 genes, but 544 remained elevated. Future studies should focus on these genes for the identification of targets that could be used in combination with MEK/ERK inhibition to increase therapeutic efficacy.

KMP is a life-threatening complication with high morbidity and mortality up to 30%[17,19]. Death usually occurs from life-threatening hemorrhage or cardiac failure. KMP affects patients with KHE and TA[19,36], and is characterized by severe thrombocytopenia and consumptive coagulopathy. While the discovery of *GNAQ/11/14* mutations in vascular anomalies is recent, a few studies have already reported an association between KMP and gain-of-function *GNA11* or *GNA14* mutations in patients[3,27]. To date, no murine models have been described that replicate the clinical characteristics of KMP, preventing investigation of the cellular and molecular mechanisms regulating the development of this coagulopathy. As a consequence, there is only one proposed cellular mechanism for the association of vascular anomalies with thrombocytopenia, which is the trapping of platelets in malformed vessels[37]. Our *iCdh5-GNAQ*$^{Q209L}$ model recapitulates several important features of KMP such as thrombocytopenia, anemia, elevated D-dimer, and accumulation of platelets and fibrinogen in vascular tufts. This local trapping of platelets and fibrinogen consumption could lead to increased risk of bleeding. Our study phenocopied important aspects of *GNAQ*-related vascular tumors and suggests that treatment with MEK/ERK inhibitors could prevent or slow the onset of coagulopathy. EC are crucially involved in vascular hemostasis. In fact, EC maintain blood fluidity by providing an anticoagulant and antithrombogenic boundary layer and by producing regulators of platelet activity. In our transcriptomic data in EC, we showed that coagulation and complement pathways were among the top differentially expressed gene sets in cultured iEC *GNAQ*-Q209L compared to iEC *GNAQ*-WT. These findings strongly suggest that transcriptional changes within the mutant *GNAQ* EC could be driving the hemostatic changes

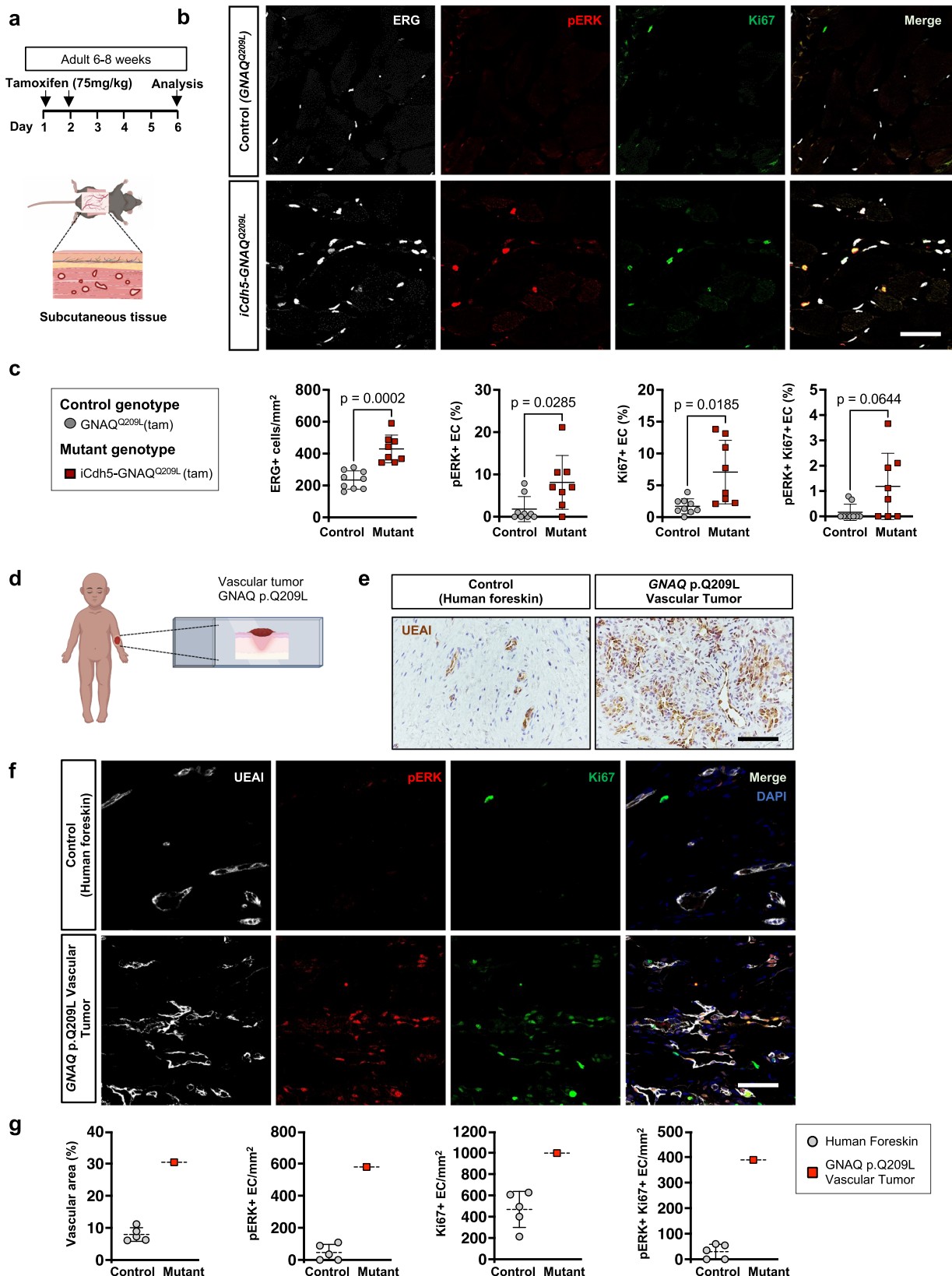

associated with KMP. By transcriptomic analysis, we have also shown that Trametinib can restore the upregulation of genes implicated in complement and coagulation directly on the ECs. Trametinib can also inhibit platelet MEK making it plausible that KMP resolution is facilitated by a dual effect on EC and platelets[61]. However, further studies are needed to dissect the cell-specific effects of Trametinib on KMP.

Finally, our study also has a few limitations. 1. Here we used mutant $GNAQ^{Q209L}$ to model common pathological features in tumors with GNA -Q, −11, and −14 hyperactive mutations. While these Gαq gene family members share about 90% sequence homology, future investigations could uncover phenotypical and signaling differences related to the specific mutated gene. 2. While in patients, lesions are visible

**Fig. 6 | Increased ERK activation in the vasculature of *GNAQ^Q209L^* expressing mice and patient vascular tumor. a** Tamoxifen induction scheme for analysis of subcutaneous tissue. **b** Representative images of subcutaneous tissue of tamoxifen-treated control and *iCdh5-GNAQ^Q209L^* mutant mice stained for ERG (white), pERK (red), Ki67 (green), and merge. Scale bar: 50 µm. **c** ERG staining marks EC nuclei. Quantification of ERG⁺ cells normalized per subcutaneous tissue area. pERK⁺ and Ki67⁺ and double-positive endothelial cells were normalized to total EC number. *iCdh5-GNAQ^Q209L^* (n = 9) and control littermates (n = 9), mean±SD, unpaired two-tailed Welch's t-test. Mouse genotypes are indicated. **d** *GNAQ^Q209L^* mutant patient-derived vascular tumor tissue was analyzed. **e** Representative images of neonatal human foreskin and patient-derived tissue (cutaneous vascular tumor) with somatic *GNAQ* p.Q209L mutation immuo-stained for UEAI (Ulex europaeus agglutinin-I) (brown). Scale bar: 50µm. **f** Representative images of neonatal human foreskin and patient-derived tissue with somatic *GNAQ* p.Q209L mutation labeled for UEAI (white), pERK (red), Ki67 (green), and merge with DAPI (blue). Scale bar: 50 µm. **g** Quantification of the UEAI⁺ vascular area as percentage of total field area, pERK⁺, Ki67⁺ and double-positive EC calculated as events per mm² of vascular area; human foreskin tissue from n = 5 different donors, and n = 1 patient-derived tumor tissue, mean±SD. Source data for (**c,g**) are provided as a Source Data file. Schematic in (**a,d**) created with Biorender.com.

and often disfiguring, in our murine models they could be detected only after dissection. Measurable superficial skin lesions did not form presumably because widespread vascular defects caused early lethality. For this reason, we used the subcutaneous and intestinal muscle tissue as model tissues to evaluate changes in the vasculature before a superficial visible lesion formed. Future studies should focus on local tamoxifen delivery to better mimic the somatic nature of the events occurring in patients. Furthermore, different doses of tamoxifen could be used to vary the number of cells that undergo CreER^T2^-mediated recombination and turn on *GNAQ^Q209L^* expression, influencing the severity of the resulting phenotype. 3. In addition, as discussed above, while we expect MAPK/ERK to be an important determinant of the vascular phenotypes (hyperproliferation and vascular permeability), our results do not exclude the involvement of other important pathways.

In conclusion, our results confirm that *GNAQ* plays an essential role in vascular development and homeostasis. The study of its mutations and molecular interactions is of great interest because it may provide new therapeutic targets to prevent the progression of vascular anomalies as well as the potentially lethal complication of KMP. The model we report here should be instrumental for testing novel targeted therapeutic strategies for the treatment of patients affected by *GNAQ*-related vascular tumors.

## Methods
### Mouse models
Mice were cared for in accordance with the National Institutes of Health guidelines, and all procedures have been reviewed and approved by the CCHMC Institutional Animal Care and Use Committee (Protocol number IACUC 2020-0039). To study the effects of constitutive active *GNAQ* expression in the developing and adult vasculature, we utilized different breeding strategies. Both male and female mice were included in the study.

**CMV-Cre; GNAQ^Q209L^ mice (CMV-GNAQ^Q209L^).** We examined the effects of *GNAQ^Q209L^* expression during embryonic development by crossing the *Rosa26-floxed stop-GNAQ^Q209L^[13]* mice with the ubiquitously expressed *CMV-Cre* (Jax stock No: 006054)[14]. Offspring (postnatal) and embryos (E.8.5 and E.13.5) were genotyped and Mendelian ratios were calculated.

**Cdh5-iCreER^T2^; GNAQ^Q209L^ mice (iCdh5-GNAQ^Q209L^).** The endothelial-specific, tamoxifen-inducible Cre-driver line *Cdh5 (PAC)-iCreER^T2^[15]* (*iCdh5*) was crossed with the *Rosa26-floxed stop-GNAQ^Q209L^ (Gt(ROSA) 26Sor^tm1(GNAQ)*Cur13^)* mouse (*GNAQ^Q209L^*). Pups received intragastric tamoxifen injection at P1 (10 µg). Adult mice (6-8 weeks) received 75 mg/kg or 40 mg/kg of tamoxifen intraperitoneally (i.p.). Tamoxifen-injected littermates (genotype: *WT, Cdh5-iCreER^T2^, GNAQ^Q209L^*) and no-tamoxifen mutant mice were used as controls. The vascular phenotype was analyzed for vascular abnormalities in subcutaneous tissue and intestinal *muscularis*.

**Pdgfb-iCreER^T2^; GNAQ^Q209L^ mice (iPdgfb-GNAQ^Q209L^).** *Pdgfb-iCreER^T2^* is a widely used transgenic mouse line that expresses a tamoxifen-inducible form of Cre-recombinase (iCreER^T2^)[16] in endothelial cells. *Pdgfb-iCreER^T2^[16]* mice were crossed with the *Rosa26-floxed stop-GNAQ^Q209L^* mouse. Pregnant dams received tamoxifen i.p. 15 mg/kg and/or pups received intragastric tamoxifen injection at P1 (1 µg). Adult mice (6-8 weeks) received 75 mg/kg of tamoxifen i.p.. Tamoxifen-injected littermates (genotype: *WT, Pdgfb-iCreER^T2^, GNAQ^Q209L^*) were used as controls. The vascular phenotype was analyzed for vascular abnormalities in brain, retina, subcutaneous tissue and intestinal *muscularis*.

**Pdgfb-iCreER^T2^; hM3Dq mice (iPdgfb-hM3Dq).** We used *Gαq-DREADD* mouse[24] that expresses a modified M3 muscarinic receptor (hM3Dq)[25]. This mouse line, *CAG-LSL-Gq-DREADD* (Jax Stock No: 026220)[25] was crossed with *Pdgfb-iCreER^T16^*, to induce Gαq activity specifically in the endothelial cells (EC). Pups were injected with tamoxifen (10µg) at postnatal day 1 and 2 and with CNO (5µg/g) daily from P3-P8. The vascular phenotype was analyzed for vascular abnormalities in the skin.

**Cdh5-iCreER^T2^; Tdtomato mice (iCdh5-tdTomato).** The *Rosa26-floxed stop-tdTomato* reporter mouse was crossed with *Cdh5-iCreER^T2^* to generate a mouse with EC-specific tdTomato expression (*iCdh5-tdTomato*). Pups received intragastric tamoxifen injections at P1 (10µg) and subcutaneous tissue was analyzed at P4. Adult mice (6-8 weeks) were tamoxifen injected (i.p. 75 mg/kg) on 2 subsequent days and analyzed at day 6 after first tamoxifen injection for vascular phenotype in subcutaneous and intestinal *muscularis* tissue. Whole-mount staining and tissue sections were analyzed for CD31 and tdTomato expression. *iCdh5-tdTomato* mice injected with sunflower oil only (no tamoxifen) were used as controls.

### Mouse husbandry
Mice were housed in the animal care facility of CCHMC under standard pathogen-free conditions with a 14 h light/10 h dark schedule and provided with food (LabDiet, #5010) and water ad libitum, temperature 22 °C and ~40% relative humidity. All mice were maintained on a C57BL/6 background and both male and female mice were used in all experiments. For survival studies of *iCdh5-GNAQ^Q209L^* mice (75 mg/kg or 40 mg/kg of tamoxifen schemes) female and male mice data was analyzed separately and did not show significant differences between the two groups. Therefore, in subsequent studies, data from female and male mice were pooled together. Mice were genotyped using EconoTaq Plus green 2X Master Mix (Lucigen). Genotyping primers are indicated in Supplementary Table 2. Birth was defined as postnatal day 0 (P0). The Cre-LoxP system was activated through tamoxifen (dissolved in sunflower oil) injections. All animals were monitored once a day for changes in their health conditions. According to guidelines established in our IACUC protocol, mice were humanely euthanized upon signs of moribundity, including lethargy, respiratory distress, and neurological defects. For endpoint studies, mice were euthanized at the established timepoint by carbon dioxide inhalation and cervical dislocation. For studies in which mice were perfused prior to tissue collection, anesthetized mice were perfused transcardially with HBSS through the left ventricle for 10 min. Macroscopic images of

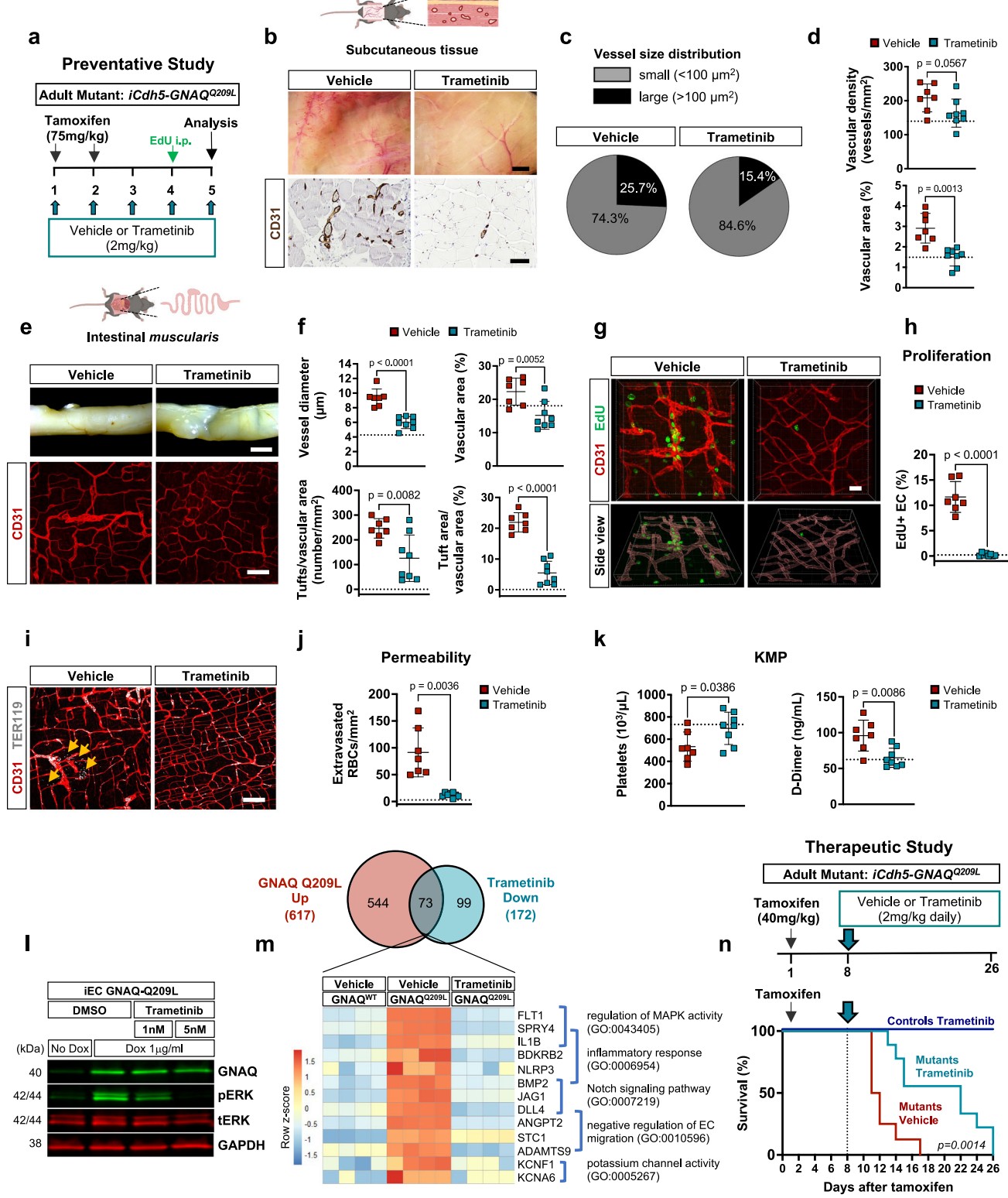

subcutaneous and intestinal tissues were taken with a Leica S8 APO stereomicroscope equipped with a Flexacam C3 camera (Leica).

The specific MEK1/2 inhibitor Trametinib (GGSK-1120212, LC laboratories, cat#T8123) was dissolved in DMSO (5 mg/mL, Sigma, Cat# D8418). Vehicle solutions for oral dosing were prepared by adding Polyethylene glycol 300 (4 mL, Sigma, cat# 807484), Tween 80 (0.5 mL, Fisher Scientific, cat# BP338-500), and normal saline (14.5 mL, Baxter, cat#2F7124). Mice were administered vehicle or Trametinib (2 mg/kg) via daily gavage.

**Patient tissue samples**

The study was performed in accordance with the Declaration of Helsinki, and the patient tissue sample was obtained after written informed consent. This study used samples, data, and/or services from the Discover Together Biobank at Cincinnati Children's Research Foundation.

All the procedures were approved by the Institutional Review Board according to ethical guidelines (Approved IRB # 2016-3878 and # 2017-3726 per institutional policies) at Cincinnati Children's Hospital Medical Center (CCHMC), with approval of the Committee on Clinical

**Fig. 7 | Trametinib treatment of *GNAQ^Q209L* mutant mice rescues vascular phenotype, hyperproliferation, permeability and coagulopathy. a** Preventative study scheme. 6 weeks-old *iCdh5-GNAQ^Q209L* mice induced with 75 mg/kg tamoxifen and concomitantly treated with Trametinib (2 mg/kg/day) or vehicle. **b** Macroscopic photographs of subcutaneous tissue from vehicle and Trametinib -treated mice. CD31 (dark brown) stained tissue sections. Scale bars: 2 mm and 50 μm. **c** Vessel size distribution in subcutaneous tissue as percentage of small (<100 μm²) and large (>100 μm²) vessels. **d** Vascular density and area in vehicle or Trametinib -treated mice. **e** Representative photograph of intestine from vehicle and Trametinib -treated *iCdh5-GNAQ^Q209L* mice. Whole-mount intestinal *muscularis* stained for CD31 (red), z-stack maximum intensity projection. Scale bars: 2 mm and 100 μm. **f** Quantification of intestinal *muscularis* vessel diameter, vascular area, tuft number and tuft area. **g** Intestinal whole-mounts labeled for EdU (green) and CD31 (red). Scale bar: 20 μm. **h** EdU positive EC (EdU⁺/CD31⁺) as percentage over total EC number (CD31⁺). **i** Representative z-stack maximum intensity projection of intestinal *muscularis* stained for TER119 (white) and CD31 (red). Yellow arrows indicate extravasated erythrocytes. Scale bar: 100 μm. **j** Quantification of extravasated erythrocytes normalized to tissue area. **k** Platelet counts and D-Dimer levels in vehicle or Trametinib -treated *iCdh5-GNAQ^Q209L* mice. In **d**, **f**, **h**, **j** and **k**: vehicle (*n* = 7), Trametinib (*n* = 8), mean±SD, unpaired two-tailed Welch's t-test. Black dotted lines indicate reference values in untreated control mice. **l** Representative immunoblot of iEC *GNAQ*-Q209L treated with DMSO or Trametinib and probed for GNAQ, phosphor-ERK (pERK, Thr202/Tyr204), total ERK (tERK), and GAPDH (*n* = 3 independent experiments). **m** Venn diagram of genes upregulated in iEC *GNAQ*-Q209L and downregulated with Trametinib treatment (6 h). Expression levels of 73 genes were normalized by Trametinib (adjusted *P*-value of <0.05 and a log2 (fold change) >1). Heatmap of selected genes clustered by GO-terms. Wald test for hypothesis testing. **n** Therapeutic study scheme. 6 weeks-old *iCdh5-GNAQ^Q209L* mice induced with 40 mg/kg tamoxifen and after 8 days treated daily with Trametinib (2 mg/kg) or vehicle. Kaplan-Meier survival curve of vehicle (*n* = 8, red) and Trametinib -treated (*n* = 9, turquoise) *iCdh5-GNAQ^Q209L* mice. Gehan-Breslow-Wilcoxon test. Trametinib-treated control mice (*iCdh5* mice, *n* = 3, blue). Source data for (**c,d,f,h,j,k,l,m,n**) are provided as a Source Data file. Schematic in (**b,e**) created with Biorender.com.

Investigation. Samples were obtained without identifiers and include excised tumor tissue sections and neonatal foreskin for immunohistochemistry and immunofluorescent staining. We do not have information on the sex/gender of the human subject of the tumor sample in the study as the specimen was deidentified. Neonatal foreskin is routinely collected from newborn males.

### Immunohistochemistry of paraffin-embedded tissue

Patient-derived tumor tissue and subcutaneous tissue and muscle of the abdomen of adult mice was fixed in 4% paraformaldehyde (PFA, Electron microscopy Sciences, cat#15710), embedded in paraffin, and sectioned at 5 μm for staining. Slides were deparaffinized with xylene and rehydrated through a descending ethanol series. Antigen retrieval was performed by boiling the slides in 0.01 M citric acid (pH 6.0). Slides were blocked overnight at 4 °C using 5% bovine serum albumin (BSA, Sigma, cat#A7906) in 0.1 M phosphate buffer saline (PBS, Fisher Scientific, cat#BP3994) containing 0.3% Triton X-100 (Sigma, cat#X100). To visualize vasculature, mouse tissue specimens were incubated overnight with the following primary antibodies: rabbit anti-CD31 (Cell Signaling, cat#77699, clone D8V9E, 0.062 μg/mL) or rabbit anti-mouse fibrinogen antiserum[62], followed by incubation with biotinylated anti-rabbit antibody (Vector laboratories, cat#BA-1000, 7.5 μg/mL) at RT for 2 h. Human specimens were incubated with biotinylated UEA-I (Vector laboratories, cat#B-1065-2, 20 μg/mL) diluted in PBS containing 5% BSA for 1 h at room temperature (RT). Peroxidase was quenched using 3% hydrogen peroxide $H_2O_2$ (Sigma, cat#H1009), followed by a 2 h incubation at RT with Horseradish Peroxidase (HRP) Streptavidin (Vector laboratories, cat#SA-5004-1, 5 μg/mL) and diaminobenzidine (DAB) (Vector laboratories, cat#SK-4100). The slides were counterstained with hematoxylin (Vector laboratories, cat#H-3401-500). For vascular area and vascular density analysis eight high-power field images (20x) were taken randomly per section using Nikon Eclipse CiS microscope, followed by vascular density (vessels/mm²) and vascular area (%) quantification with FIJI software (v1.54b, National Institutes of Health (NIH), Bethesda)[63]. Further antibody details are listed in Supplementary Table 3.

### Immunofluorescent staining and quantification of tissue sections

For immunofluorescent triple labeling, sections were blocked using 5% BSA in PBS containing 0.5% Triton X-100 overnight at 4 °C. Next, specimens were incubated with rabbit anti-pERK 1/2 (Phospho-p44/42 MAPK Thr202/Tyr204, Cell signaling, cat#9101, 5 μg/mL) antibody at 4 °C overnight. Subsequently, sections were incubated for 2 h at RT with a biotinylated anti-rabbit antibody (Vector laboratories, cat#BA-1000, 7.5 μg/mL) and Texas Red® Streptavidin (Vector laboratories, cat#SA-5006-1, 5 μg/mL) for 2 h at RT. Next, sections were incubated with Alexa Fluor™ 488 conjugated (A488) anti-Ki67 (Cell Signaling, cat#11882, clone D3B5, 2 μg/mL) overnight at 4 °C. Subsequently, slides were incubated with anti-ERG-A647 (Abcam, cat#ab196149, clone EPR3864, 5 μg/mL, mouse) or UEA-I DyLight™ 649 (Vector laboratories, cat#DL-1068-1, 20 μg/mL, human), at 4 °C overnight and counterstained with 4′,6-Diamidino-2-Phenylindole, Dihydrochloride (DAPI, cat#D1306, Invitrogen, 5 μg/mL) for 5 min at RT. Sections were mounted with Fluoromount-G® mounting medium (Southern Biotech, cat#0100-01). For quantitative analysis of mouse tissues, five randomly assigned regions were imaged per section using Nikon Eclipse T1 microscope. Image analysis and quantification were performed using the open-source software CellProfiler[64]. A CellProfiler pipeline was designed to detect and count ERG, pERK and Ki67-positive cells from immunofluorescence images using automatic thresholding and segmentation methods. For mouse tissue, cells identified as double or triple positive were quantified and expressed as percentage of the number of ERG-positive EC. For quantitative analysis of patient-derived tissue, 5-11 randomly assigned regions were imaged per section using Nikon Eclipse T1 microscope. Double and triple-positive cells were counted and normalized to UEA-I-positive vascular area using FIJI software (v1.54b, NIH). Further antibody details are listed in Supplementary Table 3.

### In vivo cell proliferation assay

To evaluate cell proliferation in vivo, we visualized the incorporation of 5-ethynyl-2′-deoxyuridine (EdU), detecting cells undergoing DNA replication. EdU was injected intraperitoneally at 50 mg/kg, 24 h before tissue harvest. EdU incorporation was detected using the click-iT EdU imaging kit (Life Technologies, cat#C10337) according to the manufacturer's instructions. The number of EdU+ nuclei within the CD31 + vessels were counted and expressed as a percentage of the total number of EC nuclei.

### Intestinal *muscularis* whole-mount preparations

For intestinal *muscularis* preparations, the gastrointestinal tract (GIT) was dissected, and the mesenteries were removed. The GIT was straightened and fixed in 10% formalin overnight at 4 °C. Approximately 2 cm of the jejunum was opened longitudinally, and the muscle layer was carefully separated from the underlying submucosa and mucosal tissue under a dissecting microscope using watchmaker forceps. Tissue was stored in PBS at 4 °C until it was processed for whole-mount immunofluorescent staining.

### Whole-mount immunofluorescence staining and vascular analysis

Specimens for whole-mount staining (subcutaneous tissue of post-natal mice and intestinal muscle tissue) were permeabilized and

blocked in 5% BSA containing 0.3% Triton X-100 in PBS for 2 h at RT. Tissues were incubated overnight at 4 °C with primary rat anti-mouse CD31 antibody (BD Biosciences, cat#550274, clone MEC 13.3, 0.075 µg/mL). Subsequently, tissues were incubated for 4 h with biotinylated goat anti-rat antibody (Vector laboratories, cat#BA-9401, 7.5 µg/mL) at RT. Biotinylated secondary antibody was detected by incubation with Texas Red® Streptavidin (Vector laboratories, cat#SA-5006-1, 5 µg/mL) for 2 h at RT. For erythrocyte extravasation analysis, tissue was incubated overnight at 4 °C with A647-conjugated anti-mouse TER119 antibody (Biolegend, cat#116218, clone TER-119, 5 µg/mL). For platelet accumulation assessment, tissue was incubated overnight at 4 °C with PE-conjugated CD41 (BD Biosciences, cat#558040, clone MWReg30, 1 µg/mL) or rabbit anti-CD42b (Abcam, cat#183345, clone SP219, 0.3 µg/ml). Subsequently, tissues were incubated for 4 h with goat anti-rabbit A594 antibody (Invitrogen, cat#A32740, 8 µg/mL) at RT. For nuclear counterstaining, tissues were incubated with DAPI (Invitrogen, cat#D1306, 5 µg/mL) for 10 min at RT. Specimens were mounted with Fluoromount-G® mounting medium (Southern Biotech, cat#0100-01). Further antibody details are listed in Supplementary Table 3.

## Quantification and visualization of the vasculature in whole-mount preparations

For quantification of the vasculature, z-stack images were acquired with Nikon Elements software on a Nikon A1R laser-scanning confocal microscope. A minimum of 3-8 randomly assigned regions were imaged per animal. FIJI (v1.54b, NIH) software was used to reconstruct the Z series as maximum intensity projection. The vascular area was quantified using Angiotool (NIH software)[65] and referred as percentage of the respective total tissue surface area. For vascular tuft analysis, vascular tufts with an area≥200µm$^2$ were outlined using FIJI software and the number of tufts as well as tuft area was quantified relative to the total vascular area. To assess blood vessel diameter, a grid was placed over each image. Vessel diameter was measured using FIJI software at locations where grid lines intersected a CD31$^+$/IB4$^+$ vessel. Three-dimensional reconstructions were made using Imaris software (Bitplane, v9.8).

## Visualization and analysis of mouse brain vasculature

For visualization of brain vascular lumina, 70kD A594-conjugated dextran (Invitrogen) was warmed to 37 °C and perfused through the heart of deeply anaesthetized mice. Subsequently, animals were perfused transcardially with ice-cold PBS, then 4% paraformaldehyde (PFA) and each mouse brain was then rapidly extracted. Tissue was cleared using PACT (passive clarity technique)[66] and embedded in RIMS solution for 3D imaging. Specimens were imaged using two-photon microscopy (Nikon A1R MP) and reconstructed using Imaris software (Bitplane). For quantitative analysis, cryosections (50 µM) were labelled using A488-IB4 (1 µg/mL; Invitrogen) at 4 °C overnight and counterstained with DAPI for 5 min at RT. Sections were mounted with Fluoromount-G® mounting medium (Southern Biotech) and Z-stack images were acquired on a Nikon A1R laser scanning confocal microscope. FIJI software was used to reconstruct the Z series as maximum intensity projection. Vascular tufts (area≥ 200µm$^2$) were outlined, and the tuft area was quantified relative to the total vascular area.

## Retina and hyaloid whole-mount preparation

Eyes from P8 mice were enucleated and fixed in 4% PFA on ice. Retina whole-mount preparations were generated[67–69] and labeled with the following: A488-IB4 and DAPI (1 µg/mL; Invitrogen).

## Hematologic parameters

Blood was collected from the inferior vena cava (IVC) of adult mice. The abdominal cavity was opened and approximately 500µL of blood was drawn in a 27-gauge syringe pre-loaded with 50uL of 0.105 M sodium citrate as anticoagulant. Samples were analyzed for blood counts using a Hemavet 950 instrument (Drew Scientific). To obtain plasma, samples were centrifuged at 1100 G for 10 min at RT and plasma was stored at −80 °C. Plasma D-dimer (Asserachrom® D-Di, Diagnostica Stago, cat# NC9884012) levels were determined by enzyme-linked immunosorbent assays following manufacturer's instructions. Blood smears were prepared via the 'push' (wedge) method by placing a small drop (4-6µL) of blood on a slide, spreading the drop using another slide at a 30-45° angle to create a thin smear with a feathered edge, and allowing the smears to air dry for at least 30 min. They were then Wright-Giemsa stained with the Bayer HEMA-TEK2000 slide stainer and imaged at 100x with oil immersion. Between 5-9 photos were taken per mouse with schistocytes and polychromasia quantified per high power field and averaged.

## Bone marrow analysis

Bone marrow (BM) cells were obtained after BM was flushed, treated with red blood cell lysis buffer (150 mM NH$_4$CL and 10 mM KHCO$_3$) and washed with staining media (Hank's Buffered Salt Solution supplemented with 2% fetal bovine serum)[39]. $8 \times 10^6$ unfractionated BM cells were stained with unconjugated rat lineage-specific antibodies (Ter119, Mac1, Gr-1, B220, CD5, CD3, CD4, CD8) followed by staining with goat anti-rat PE-Cy5 antibody. Cells were then stained using c-Kit-APC-eFluor780, Sca1-PB, CD48-BV711, CD150-PE, Flt3-Biotin, FcgR-PerCP-eFluor710, CD34-FITC, CD41-BV605, and CD105-APC antibodies. Secondary staining was performed with streptavidin-PE-Cy7. Zombie NIR fixable viability kit (BioLegend) was used for dead cell exclusion. Data were collected on a 5 laser Aurora spectral flow cytometer (Cytek Biosciences). Data analysis was performed using FlowJo (BD biosciences) software. Further antibody details are listed in Supplementary Table 5.

## Miles assay

For in vivo permeability assay, the dorsal flank of adult mice (6−8 weeks) was bilaterally shaved 24 h prior the experiment. The next day, the histamine H1 receptor antagonist pyrilamine maleate salt (4 mg/kg body weight in 0.9% saline) was injected i.p. 30 min before Evans blue injection to block the effects of local histamine release due to injection-induced mast cell activation. Evans blue (100 µL of 1% in sterile saline) were injected into the tail vein and allowed to circulate for 20 min. VEGF (100 ng/50 µL, R&D Systems) and PBS were injected intradermally into the dorsal flanks. Twenty minutes later, mice were sacrificed by cervical dislocation and the dorsal skin was excised. The weight of the excised skin tissue was noted, and the samples were dried overnight by placing them into 1.5 mL tubes inside a heating block at 55 °C. Evans blue was extracted from the excised tissue by immersion in formamide for 24 h at 55 °C and the amount of blue dye was quantified by spectrometry at 620 nm. The measured OD of each sample was normalized by the weight of the excised tissue and shown as change (ΔOD620) of VEGF-A-treated tissue relative to PBS-treated tissue.

## Lentiviral vector

Full-length wild-type human *GNAQ* cDNA was purchased from Sino Biologicals (cat# HG17607-U, Wayne, PA) in a pUC19 cloning vector. The specific mutation in c.626 A > T (p.Gln209Leu), was introduced by site-directed mutagenesis using the Q5® Site-Directed Mutagenesis Kit (New England BioLabs Inc., cat#E05545) (Supplementary Fig. 19). The following primers were used to introduce the c.626 A > T mutation: Forward 5'-GTAGGGGGCCtAAGGTCAGAG −3', and Reverse 5'- ATC GACCATTCTGAAAATGACAC −3'. The *GNAQ* WT and Q209L cDNA was subsequently introduced into the lentiviral vector pCW-Cas9 (pCW-Cas9 was a gift from Eric Lander & David Sabatini (Addgene plasmid # 50661; http://n2t.net/addgene:50661; RRID:Addgene_50661), at NheI(5') and BamHI I (3') to replace the Cas9 gene. The lentivirus was

generated by using second-generation packaging system at the Cincinnati Children's Hospital viral vector core facility.

## Lentiviral transduction of human endothelial cells

Human endothelial colony-forming cells (ECFC, StemBiosys) were plated onto fibronectin-coated (1 μg/cm², Sigma, cat#FC010) plates and cultured in endothelial growth medium (EGM2, Lonza, cat# CC-3162) supplemented with 10% fetal bovine serum (FBS) (GE Healthcare, cat#SH30910.03) and 1% penicillin-streptomycin-glutamine (Corning, Cat#30-009-CI). Cells were treated with Hexadimethrine bromide (8 μg/mL, Sigma, cat# TR-1003-G) and lentiviral particles containing pCW-*GNAQ*-WT or pCW-*GNAQ*-Q209L were added. After 24 h the media was replaced with puromycin (1 μg/mL, Gibco, cat#A1113803) containing media to select for cells that had taken up the constructs. Lentiviral-engineered ECFC expressing WT or Q209L *GNAQ* cDNA were designated as iEC *GNAQ*-WT and iEC *GNAQ*-Q209L, respectively. Doxycycline (Dox, Sigma, cat#D3447) was added to the media to induce *GNAQ*-WT or *GNAQ*-Q209L expression (0.25-2 μg/mL) for 48 h before cells were lysed for Western Blot analysis. For inhibitor studies, cells were treated with 1 nM, 5 nM or 10 nM Trametinib or vehicle (DMSO 1 μL/mL) for 20 min.

## Immunofluorescence monolayer staining for VE-cadherin

Immunofluorescence staining was performed on 4% PFA fixed ECFC monolayers with an A647-conjugated VE-Cadherin antibody (BD Biosciences, cat#561567, 0.5 μg/mL), for 2 h at RT. For nuclear counterstaining, cells were incubated with DAPI (Invitrogen, D1306, 5 μg/mL) for 10 min at RT. Specimens were mounted with Fluoromount-G® mounting medium (Southern Biotech, cat#0100-01). Z-stack (10 μM) confocal images were acquired on a Nikon A1R laser scanning confocal microscope. For quantification, VE-Cadherin positive area was measured using FIJI software (v1.54b, NIH) and normalized to the number of cells. Cells from *n* = 5 independent experiments (for each experiment *n* = 10 images) were used for quantifications.

## Immunoblotting

Cells were washed with PBS then lysed using ice-cold RIPA lysis buffer (Boston Bioproduct) supplemented with HALT™ protease/phosphatase inhibitor cocktail (Thermo Scientific, cat# 78442). The protein concentration was determined using the BCA Protein Assay Kit (Thermo Scientific, cat# PI23225). 20 μg of total protein were subjected to SDS-PAGE (4-20%, Midi Criterion precast gels, Bio-Rad, cat#5678094) and transferred to a PVDF membrane (Immobilon®-P PVDF Membrane, Millipore, cat#IPVH00010). Membranes were blocked for in 5% nonfat dried milk for 1 h at RT and probed with the following antibodies: rabbit anti-GNAQ (Cell Signaling, cat#14373, clone D5V1B 0.05 μg/mL), mouse anti-VE-Cadherin (Santa Cruz, cat# sc-9989, clone F-8, 0.5 μg/mL), rabbit anti-pERK 1/2 (Thr202/Tyr204, Cell Signaling, cat#9101, polyclonal, 0.5 μg/mL), mouse anti-ERK 1/2 (Cell Signaling, cat#4696, clone L34F12, 0.5 μg/mL), rabbit anti-pAKT (Ser473, Cell Signaling, cat#4060, clone D9E, 0.5 μg/mL) and mouse anti-AKT (Cell signaling, cat#2920, clone 40D4, 0.2 μg/mL), goat anti-Angiopoietin-2 (R&D Systems, cat#AF-623, polyclonal, 1 μg/mL) and mouse anti-GAPDH (Millipore, cat# MAB374, clone 6C5, 0.2 μg/mL). Membranes were incubated with the following secondary antibodies: Donkey anti-mouse IgG-DyLight™ 680 (Invitrogen, cat# SA5-10170, 0.1 μg/mL), goat anti-rabbit IgG-DyLight™ 800 (Invitrogen, cat#SA5-10036, 0.1 μg/mL) or donkey anti-goat IgG-DyLight™ 800 (Invitrogen, cat#SA5-10092, 0.1 μg/mL). Antigen–antibody complexes were visualized using Odyssey Scanner and analyzed using Image Studio Software (LI-COR, v5.2). Further antibody details are listed in Supplementary Table 3.

## RNA isolation and RNA sequencing

Cells were cultured in 2% FBS EGM2 containing 1 μg/mL doxycycline for 18 h. Cells were treated with Trametinib (10 nM, LC laboratories,

cat#T-8123) or vehicle for additional 6 h. Total RNA was extracted using RNeasy extraction kit (Qiagen) according to manufacturer's recommendations. RNA integrity number and concentration was assessed using a 5300 Fragment Analyzer System (Agilent) at the DNA Sequencing and Genotyping Core. 300 ng of total RNA was used for library preparation using the Illumina Stranded mRNA Prep–Ligation kit. The libraries were then quantified and sequenced using an Illumina NovaSeq 6000 at a sequencing depth of 40 million reads per sample. RNA-seq reads in FASTQ format were first subjected to quality control to assess the need for trimming of adapter sequences or bad quality segments. The programs used in these steps were FastQC v0.11.7 (http://www.bioinformatics.babraham.ac.uk/projects/fastqc), Trim Galore! v0.4.2 (https://www.bioinformatics.babraham.ac.uk/projects/trim_galore) and Cutadapt v1.9.1[70]. The trimmed reads were aligned to the reference human genome version hg38 with the program STAR v2.6.1e[71]. Aligned reads were stripped of duplicate reads with the program Sambamba v0.6.8[72]. Gene-level expression was assessed by counting features for each gene, as defined in the NCBI's RefSeq database[73]. Read counting was done with the program featureCounts v1.6.2 from the Rsubread package[74]. Raw counts were normalized as transcripts per million (TPM). Differential gene expressions between groups of samples were assessed with the R package DESeq2 v1.26.0[75]. Gene list and log2 fold change are used for GSEA analysis using GO pathway dataset[76,77]. Plots were generated using the ggplot2 v3.3.6[78] package and base graphics in R. Heatmaps were created using pheatmap (RRID:SCR_016418, v1.0.12.). Gene ontology (GO) and Kyoto Encyclopedia of Genes and Genomes (KEGG) pathway analyses were performed using ShinyGO (V.0.76.2)[79] where adjusted *p* value (FDR) cut-off was set on 0.05. RNA-seq data was submitted to the Gene Expression Omnibus (GEO) (Accession number: GSE216367).

## Real-time reverse transcriptase PCR

Reverse transcriptase reactions were performed using an iScript cDNA synthesis kit (Bio-Rad, cat#1708841). qPCR was performed using SsoAdvanced Universal SYBR(R) Green Supermix (Bio-Rad, cat#1725272). Amplification was performed in Bio-Rad Touch Real-time PCR detection system (CFX96). A relative standard curve of each gene amplification was generated and an amplification efficiency of >90% was considered acceptable. Hypoxanthine phosphoribosyl transferase 1 (*HPRT1*) and TATA-binding protein 1 (*TBP1*) were used as housekeeping genes. Quantification was performed using the Pfaffl method[80]. Primer sequences are shown in Supplementary Table 4.

## Data collection and statistics

Excel (v16.67) was used to collect and organize raw data. Prism 9.0 software (GraphPad Software, v9.3.1) was used for all statistical assessments. Data analyzed statistically are presented as mean±standard deviation (SD) of two or more biological replicates (*n* values reported in figure legends). Statistical significance between two groups was assessed by parametric Welch's t-test. When more than two groups were compared, one or two-way ANOVA was used followed by Tukey, Dunnett's or Sidak's post hoc test. Differences were considered significant for *P* value less than 0.05. Schematics in all figures were created using Biorender.com.

## Reporting summary

Further information on research design is available in the Nature Portfolio Reporting Summary linked to this article.

## Data availability

Source data are provided with this paper. The generated RNA-Seq data in this study has been deposited in GEO under the accession number GSE216367. The CellProfiler pipeline can be downloaded here: https://cellprofiler.org/published-pipelines. All data generated in this study

are provided in the Source Data file and in Supplementary information file. Source data are provided with this paper.

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

## Acknowledgements

Research reported in this manuscript was supported by the National Heart, Lung, and Blood Institute, under Award Number 2R01 HL117952 (E.B.), part of the National Institutes of Health. The project described was also supported by the National Center for Advancing Translational Sciences of the National Institutes of Health, under Award Number 2UL1TR001425-05A1 (E.B.). The content is solely the responsibility of the authors and does not necessarily represent the official views of the National Institutes of Health. Additional funding supporting the study was provided by the Charlotte R. Schmidlapp Women Scholars program at Cincinnati Children's Hospital (E.B.), the American Heart Institute (AHA) under award number 833891 (S.Schrenk) and NIH U54 DK126108 to the hematology center core (CCHMC). We thank Dr. Timothy Phoenix for assistance with brain vascular phenotype analysis, Drs. Adrienne Hammill, Kiersten Ricci for discussions, Dr. Ralf Adams for the *Cdh5 (PAC)-CreER^{T2}* mice, Rachael Kang for her assistance with quantification of vascular tufts and diameter, Dr. Jorie Gatts for the analysis of blood smears. We thank the Discover Together Biobank for support of this study, as well as participants and their families. We thank the Confocal Imaging Core (CIC), Flow cytometry Core, DNA sequencing Core, Biomedical Informatics Core (SCR_022622), Veterinary Services (VET) and viral vector core facility

(VVC) at Cincinnati Children's Hospital Medical Center for providing state-of-the-art instrumentation, services, training, and education.

## Author contributions

E.B., S. Schrenk, L.J.B, J.G., Y.C., C.V.R., R.L., J.S.P. and D.R. conceived the project. E.B. supervised the research. S. Schrenk, L.J.B., J.G., D.R. and S.R.G. performed mouse studies, quantifications, data analysis, immunostainings. Y.C. performed the brain, retina and hyaloid immunostaining and initial analysis. S.V. and Y.O. performed retina dissection and initial analysis. S. Szabo performed pathology on the patient tissue. E.B. and S. Schrenk prepared the figures and the manuscript. All the authors reviewed the manuscript.

## Competing interests

The authors declare no competing interests.
