## [Peer Review File · Nature Communications]

MEK inhibition reduced vascular tumor growth and coagulopathy in a mouse model with hyperactive GNAQREVIEWER COMMENTS

Reviewer #1 (Remarks to the Author):

Schrenk et al present a novel mouse model for GNAQ-related vascular anomalies. Inducible EC-specific expression of a gain-of-function GNAQ Q209L at different developmental stages or in adult mice led to vascular lesions characterized by enlarged vessels, and increased EC proliferation. In line with previous studies in cultured ECs and melanoma cells, GNAQ Q209L expression in mouse endothelium also led to increased ERK activation. Finally, the authors show that the formation of vascular lesions could be inhibited by a treatment with a MEK inhibitor.

The novelty of this study relates to the generation and characterization of the first genetic mouse model for GNAQ-related vascular pathology. The study remains, however, descriptive and does not provide new mechanistic insight into the associated disease mechanisms. It is of interest that GNAQ mutations are causative of hemangiomas and capillary malformations. It is not clear if the model generated in this study recapitulates the pathophysiology of these two distinct disease entities but it could potentially be used to address clinically important questions on their etiology and pathogenesis.

Major comments:

1. The authors' conclusion that 'Gaq hyperactivation in EC induces vascular defects that are reminiscent of vascular anomalies such as CM/SWS' is not fully supported by data (Fig S1). They show increased vascular density based on staining of sections of facial skin, but why was the phenotype not analyzed in the retina, as done in Fig 2? The authors instead study hyaloid vessels that regress postnatally. The phenotype shown in Fig S1D could reflect lack of hyaloid vessel regression rather than increased vascularity. This data does not seem relevant to the current study.
2. The analysis of the vascular phenotype is superficial, and mainly limited to the assessment of vascular tuft area. Quantification of tuft area over total area (Fig 1D, Fig 2C) will always show significant increase in the mutant since the control group does not have any tufts. This quantification may therefore not be meaningful.
3. Are only certain vascular beds/organs or vessel types (artery/capillary/vein) sensitive to the activation of GNAQ, and does this mimic human disease pathology? Consistent analysis of different organs at different developmental stages should be provided.
4. Vascular membrane marker would be useful in Fig 3 to relate ERK activity to vessel identity (artery/capillary/vein) and anatomy (tuft/normal).
5. The authors see vascular tuft formation in adult GNAQ Q209L mice 4 days after tamoxifen injection (Fig 4). It is expected to take at least 1 day to allow for sufficient amount of mutant protein to be expressed, thus it is unexpected that a severe phenotype develops in the adult vasculature in such a short time frame. I am wondering if the vascular phenotype observed in these mice reflects tamoxifen-independent leakage leading to transgene expression and vascular expansion already at an earlier developmental stage. Cre leakage has been reported to occur in both lines used in the study (PMID: 31641921). What is the genotype of the control mice, are they GNAQ Q209L Cre⁺ mice that were not treated with tamoxifen, or Cre⁻ mice?
6. Related to the previous comment, it is also very surprising that the authors see hemostatic complications including low platelet numbers and elevated D-dimer levels in adult GNAQ Q209L just 5 days after tamoxifen injection (Fig 7). Have the authors excluded a potential targeting of and direct effect on hematopoietic cells? I believe that Kasabach-Merritt phenomenon is thought to result from consumption of platelets and fibrinogen by intralesional thrombosis. Please provide supporting evidence for intralesional thrombosis, and exclude the effect of tamoxifen-independent leakage by providing the analysis of untreated control mice carrying both GNAQ Q209L and Cre alleles.

7. Considering the potential tamoxifen-independent recombination discussed above, MEK inhibitor experiments should also include untreated control mice carrying the GNAQ Q209L and Cre alleles. Bright-field images (such as shown in Fig 4E) and/or lower magnification immunofluorescence images should be shown. In addition, it would be important to assess the effect of MEK inhibition on already established lesions (i.e. can this lead to the regression of lesions).

8. Increased pERK and KI67 staining in a human vascular tumor tissue in Fig 6 provides additional support for the involvement of ERK in GNAQ-driven pathology, and confirms previous observations (PMID: 30112971), but only 1 sample is analyzed. The comparison of vascular parameters in vascular tumor and human foreskin is not appropriate, considering that these tissues have different organization of the vasculature and represent different anatomic location. Since vascular tumors are known to be proliferative and may show increased pERK and KI67 independent of GNAQ activity, additional analysis of GNAQ-driven capillary malformations would be helpful.

Minor comments:

1. The scheme shown in Fig 1A and B is confusing. Please indicate for each experiment which Cre line and tamoxifen administration protocol was used.

2. The genotype of the control mice should be specified.

3. Some of the information is repetitive; for example, Fig 2 and 3 could be combined.

4. Line 162, 163: 'Two-photon microscopy and Imaris image analysis showed the presence of perfused vascular tufts (see arrows) in the brain (Fig.1E).' Non-perfused tufts would not be visualized, but immunofluorescence analysis of brain sections after an injection with a fixable dextran could demonstrate that all lesions are perfused.

Reviewer #2 (Remarks to the Author):

The authors established two mouse models in activating *Gaq11* or overexpressing *GNAQR183A* specifically in vascular endothelial cells. From the phenotype analyses, ERK1/2 activation and MEK inhibitor therapeutics, it was clearly demonstrated that GNAQ plays an essential role in vascular development and homeostasis.

General and major concerns: The endothelial cell-specific activation and expression were not characterized. In general, most of figures on microvessels lack high power images. Phenotypes of mice were not analyzed thoroughly, for example, the gross growth, and survival rate, and physiological changes (e.g why early lethal if it was, which was mentioned in the discussion). The work also lacks novel mechanistic insights into how GNAQ cause vascular malformations and solely selected ERK1/2 for study. Therefore, the study was considered as mouse model with phenotypic verifications.

The phenotype characterization: it was limited to a very short time frame (3-5 days after induction) without clear rationale. Any late time points were analyzed?

Some specific concerns (could be applied to all figures):

Fig.1: Two photon imaging was nice. A dynamic video would help to visualize perfusion and flow within the tufts.

Fig. 2: B/C: Quantifications of vessel density in retinas was missing.

Section imaging of retinas and hyaloid would have better visualize the tufts.

Vessel diameters: were they artery, vein or microvessels? Were microphage number differed in KO groups vs WT?

Some tissues were examined (from Fig.1-5). However, the study was simply vascular densities. No functional studies were performed.

Fig.6 had human specimens were strength.

Fig.7 therapeutics with MEK inhibitor was nice. It would be more informative to dissect why ERK1 and if GNAQ inhibitors had better efficacy in the mouse models.

Reviewer #3 (Remarks to the Author):

In this manuscript, Schrenk et al. generated the first murine model for GNAQ-related vascular anomalies, exhibiting endothelial cell (EC)-specific expression of the constitutively active mutant GNAQ pQ.209L of the Gαq protein. Gαq gain-of-function mutation results in widespread abnormal vascular morphogenesis and potential thrombocytopenia and severe coagulopathy. Such vascular morphogenesis defects are consistent with findings in patients with Gαq gain-of-function mutations. The authors characterized vascular abnormalities in the brain, retina, hyaloid, subcutaneous muscle and intestinal muscle in GNAQQ209 mice at different time points of development, and confirmed MAPK/ERK activation and proliferation markers in patient-derived tissue. Finally, the authors performed proof-of-concept experiments, demonstrating that MEK inhibition with Trametinib rescued vascular and coagulopathy phenotypes.

Since mechanisms by which Gαq regulates aberrant vascular development and remodeling remain unknown, this paper provides a murine model for studying (EC-specific) GNAQ-related vascular anomalies that will aid in development of targeted therapies, which are currently lacking. The authors provide extensive evidence of aberrant vascular morphologies that recapitulate what are seen in patients. The manuscript is easy to follow, and figures containing schematics of mouse breeding and experimental timeline are helpful. However, there are important limitations in this manuscript that limit enthusiasm that are listed below as well as more minor concerns.

MAJOR CONCERNS

1. This study is very observational. There was no further insights into the functionality of the altered vasculature. How does mutant Gαq affect EC biology beyond EC proliferation? Is the endothelium more inflamed?
2. There are very important controls not done in this manuscript that are needed to complete the claim that an EC-specific overexpression of GαqQ209 is responsible for the observed vascular effects. These include showing that there was no leakage of expression until tamoxifen was given. The controls in Figs 2-8 were done in mice that were littermates but did not have the GαqQ209 expression vector. It should have been done in mice with the GαqQ209 construct but without tamoxifen. Were the vessels normal until that was done? Also were the non-ECs normal and the effect was targeted to the endothelium? Finally, what was the relative level of GαqQ209 pre and post tamoxifen versus native Gαq? Was the effect seen just because there is a massive increase in Gαq in the ECs and even normal Gαq would have resulted in the same outcome.
3. There were two different EC-specific promoters studied. The rationale for that and why some studies were done mixing the two or with only one is not explained other than one statement that the results were not different, yet we don't know efficiency of the two in driving human GαqQ209 expression in different tissues.

MINOR CONCERNS

1. The authors should discuss Huang et al. ATVB 2022. (PMID: 34670408) on EC biology and its support of the studies in this manuscript.
2. A report of complete blood count and coagulation tests (PT and aPTT) in these mice as well as blood smears to look for schistocytes/polychromasia would be helpful. The effects of Trametinib on the blood counts should be compared to the non-tamoxifen-treated control discussed above as well as wildtype mice.
3. Page 6 – line 193: please explain the difference in subcutaneous tissue vascular area in early

onset versus adult onset of GNAQ mutation. Why is it that only in early onset of GNAQ mutation that the authors observed no change in subcu. vascular area?

4. How does mutant Gαq affect EC biology beyond cell proliferation? Is the endothelium more inflamed? Do EC express higher levels of surface adhesion molecules? What about vascular permeability? Are there any changes in blood pressure longevity?

5. Please provide evidence of thrombosis in the vasculature. For example, fibrin & platelet deposition at the lesions?

6. Trametinib also inhibits platelet MEK. Is the observed reverse of thrombocytopenia and coagulopathy seen in Trametinib-treated iEC-GNAQQ209L mice due to direct effect of Trametinib on platelets and other cells? Please discuss.

7. Consider changing the title of the manuscript? The current title states what is already known, not really what the authors are trying to show.

8. Since both male and female mice were used, did the author observe any sex difference?

9. An edited manuscript will be returned, but minor typos include: Use hyphen when appropriate.

“Ki67-postive” not “Ki67 positive”. Double check μm versus μM . Some units are mixed up.

Subcutaneous “tissue”, not subcutaneous “muscle”. Use μL , not uL . Check spacing consistency throughout. For example, $\text{area} \geq 200\mu\text{m}^2$ vs. $\text{area} \geq 200\mu\text{m}^2$. $\text{Mean} \pm \text{SD}$ vs. $\text{mean} \pm \text{SD}$.

Title: MEK inhibition reduced vascular tumor growth and coagulopathy in a mouse model with endothelial hyperactive GNAQ

Authors: Sandra Schrenk^{1,2}, Lindsay J Bischoff^{1,3}, Jillian Goines¹, Yuqi Cai¹, Shruti Vemaraju⁴, Yoshinobu Odaka^{4,5}, Samantha R Good¹, Joseph S. Palumbo^{2,6}, Sara Szabo^{2,7}, Damien Reynaud^{1,2}, Catherine D Van Raamsdonk⁸, Richard A Lang^{2,9}, Elisa Boscolo^{1,2*}

RESPONSE TO REVIEWERS

Reviewer #1 (Remarks to the Author):

Schrenk et al present a novel mouse model for GNAQ-related vascular anomalies. Inducible EC-specific expression of a gain-of-function GNAQ Q209L at different developmental stages or in adult mice led to vascular lesions characterized by enlarged vessels, and increased EC proliferation. In line with previous studies in cultured ECs and melanoma cells, GNAQ Q209L expression in mouse endothelium also led to increased ERK activation. Finally, the authors show that the formation of vascular lesions could be inhibited by a treatment with a MEK inhibitor.

The novelty of this study relates to the generation and characterization of the first genetic mouse model for GNAQ-related vascular pathology. The study remains, however, descriptive and does not provide new mechanistic insight into the associated disease mechanisms. It is of interest that GNAQ mutations are causative of hemangiomas and capillary malformations. It is not clear if the model generated in this study recapitulates the pathophysiology of these two distinct disease entities but it could potentially be used to address clinically important questions on their etiology and pathogenesis.

R: We thank the reviewer for the feedback on our study. We have extensively revised our manuscript to address the major points raised and provide novel mechanistic insights on GNAQ Q209L-related vascular tumors.

Major comments:

1. The authors' conclusion that 'Gαq hyperactivation in EC induces vascular defects that are reminiscent of vascular anomalies such as CM/SWS' is not fully supported by data (Fig S1). They show increased vascular density based on staining of sections of facial skin, but why was the phenotype not analyzed in the retina, as done in Fig 2? The authors instead study hyaloid vessels that regress postnatally. The phenotype shown in Fig S1D could reflect lack of hyaloid vessel regression rather than increased vascularity. This data does not seem relevant to the current study.

R: We agree with this comment. In this revised manuscript we have removed the hyaloid data and the link with CM/SWS. Data in the new Suppl. Fig.S3 show increased vascularity in the skin of mutant mice expressing the endothelial-specific hyperactive Gαq with the use of hM3Dq system – this data is supportive of our findings with the GNAQ^{Q209L} model.

2. The analysis of the vascular phenotype is superficial, and mainly limited to the assessment of vascular tuft area. Quantification of tuft area over total area (Fig 1D, Fig 2C) will always show significant increase in the mutant since the control group does not have any tufts. This quantification may therefore not be meaningful.

R: The vascular tufts are the main histopathological hallmark of GNAQ-related vascular anomalies, and it is important to document their number and size as this will enable preclinical studies with the use of pharmacological therapies. In this substantially revised manuscript, in addition to the tuft analysis, we have extended our characterization of the vascular defects which now includes several parameters: 1-vascular density, 2-diameter, 3-vessel size distribution, 4-number and 5-area of tufts. Furthermore, to increase the relevance of our study, we have characterized vascular permeability (Fig.3, Suppl. Video 3), coagulopathy (Fig.4, Suppl. Fig.S11-13, Suppl. Video 4), and transcriptomic analysis of the GNAQ-mutant EC (Fig.5, Suppl. Fig.S14).

3. Are only certain vascular beds/organs or vessel types (artery/capillary/vein) sensitive to the activation of GNAQ, and does this mimic human disease pathology? Consistent analysis of different organs at different developmental stages should be provided.

R: This is a good question. GNAQ-related vascular anomalies primarily affect capillaries and veins. In this revised manuscript we utilized the developing vasculature of the retina as a model in which arteries and veins can be readily distinguished. Analysis of mutant *iEC-GNAQ^{Q209L}* mice at P8 showed that vascular tufts were exclusively localized in retinal veins and capillaries, while absent in the arteries (see new Suppl. Fig.S5).

4. Vascular membrane marker would be useful in Fig 3 to relate ERK activity to vessel identity (artery/capillary/vein) and anatomy (tuft/normal).

R: We agree this would a great experiment, however pERK staining did not work in whole mounts neither in flat mount retinas. To address this point, we now show EdU staining labeling proliferative cells in whole mount intestinal *muscularis* and show that EdU+ ECs are located in the vascular tufts (Fig.2H-J and Suppl Video 2). Furthermore, we have analyzed EC proliferation in the vasculature of Trametinib-treated mice and see a significant reduction of EdU+ EC compared to vehicle-treated mice (Fig. 7G, Suppl Video 5, and Suppl. Fig. S15), confirming the importance of ERK activity in EC hyperproliferation and tuft formation.

5. The authors see vascular tuft formation in adult GNAQ Q209L mice 4 days after tamoxifen injection (Fig 4). It is expected to take at least 1 day to allow for sufficient amount of mutant protein to be expressed, thus it is unexpected that a severe phenotype develops in the adult vasculature in such a short time frame. I am wondering if the vascular phenotype observed in these mice reflects tamoxifen-independent leakage leading to transgene expression and vascular expansion already at an earlier developmental stage. Cre leakage has been reported to occur in both lines used in the study (PMID: 31641921). What is the genotype of the control mice, are they GNAQ Q209L Cre+ mice that were not treated with tamoxifen, or Cre- mice?

R: We thank the reviewer for bringing up this point so we can increase the rigor of our study. In this revised manuscript we now include no-tamoxifen mutant mice controls for most of the experiments and highlight in each graph the specific genotype of mice used for each experiment. Pups and young adults mutant mice that did not receive tamoxifen had a phenotype comparable to WT mice. Additionally, we have included comprehensive analysis of efficiency and specificity of the *Cdh5-iCreERT²* driver with and without tamoxifen, by crossing *GNAQ^{Q209L}* mice with the *Rosa26^{tdTomato}* lineage reporter. There was only minimal recombination in *Cdh5-iCreERT²; tdTomato* mice that were not administered with tamoxifen (1.8±0.88% in pups; 2.7±2.50% and 2.20±1.60% in adult mice subcutaneous and intestinal *muscularis* tissues) (see Suppl. Fig.S2 and S8). We have also analyzed survival of mutant mice that did not receive tamoxifen, and at 13 months mice are alive and do not have overt phenotype or illness.

6. Related to the previous comment, it is also very surprising that the authors see hemostatic complications including low platelet numbers and elevated D-dimer levels in adult GNAQ Q209L just 5 days after tamoxifen injection (Fig 7). Have the authors excluded a potential targeting of and direct effect on hematopoietic cells? I believe that Kasabach-Merritt phenomenon is thought to result from consumption of platelets and fibrinogen by intralesional thrombosis. Please provide supporting evidence for intralesional thrombosis, and exclude the effect of tamoxifen-independent leakage by providing the analysis of untreated control mice carrying both GNAQ Q209L and Cre alleles.

R: These are interesting questions. 1-We have included experts in the phenotypical analysis of the bone marrow HSC in this revised manuscript and demonstrate that in mutant mice there are no changes in the bone marrow cellularity neither a decrease in megakaryocyte progenitors. This suggests no involvement of the hematopoietic stem cells in the KMP phenotype. These data are included in Suppl. Fig. S13). 2- We now provide evidence of platelet trapping and intralesional fibrinogen accumulation in the vascular tufts (Fig.4, Suppl. Fig.S12 and Video 4). 3- Furthermore, we now include data on complete cell counts (CBC) including platelets and RBC, and D-dimer for no-tamoxifen mutant controls and tamoxifen-treated Cre-allele controls (see Fig. 4 and Suppl. Figs.S10).

7. Considering the potential tamoxifen-independent recombination discussed above, MEK inhibitor experiments should also include untreated control mice carrying the GNAQ Q209L and Cre alleles. Bright-field images (such as shown in Fig 4E) and/or lower magnification immunofluorescence images should be shown. In addition, it would be important to assess the effect of MEK inhibition on already established lesions (i.e. can this lead to the regression of lesions).

R: 1. To address the reviewer's question we included data on untreated (no tamoxifen) control mice carrying the GNAQ Q209L and the Cdh5-CreER^{T2} alleles in Figs.1, 2 and 4. These do not show a vascular phenotype. The potential effects of Trametinib on control mice was investigated in the therapeutic scheme (see Fig.7N) and these mice lived until the endpoint of the experiment. 2. In this revised manuscript we have included brightfield images of the subcutaneous and intestinal tissue for the Trametinib treatment experiments. 3. We have performed a therapeutic treatment scheme with Trametinib on already formed lesions. For this experiment we have used a lower dose of tamoxifen to extend the life span of the mice. Daily Trametinib treatment started 8 days after tamoxifen induction, and significantly extended the life span of the mutant mice (see Fig.7N).

8. Increased pERK and KI67 staining in a human vascular tumor tissue in Fig 6 provides additional support for the involvement of ERK in GNAQ-driven pathology, and confirms previous observations (PMID: 30112971), but only 1 sample is analyzed. The comparison of vascular parameters in vascular tumor and human foreskin is not appropriate, considering that these tissues have different organization of the vasculature and represent different anatomic location. Since vascular tumors are known to be proliferative and may show increased pERK and KI67 independent of GNAQ activity, additional analysis of GNAQ-driven capillary malformations would be helpful.

R: We have only 1 patient sample with confirmed GNAQ p.Q209L mutation and it is not possible to access control tissue from the desired anatomic location in human subjects. We think the use of neonatal foreskin from different donors should be acceptable as control as it is a proliferative and highly vascularized tissue. Furthermore, data has been carefully normalized by EC number to account for the increased EC number in the vascular tumor. We understand the Reviewer's point about the specificity of increased ERK signaling in EC expressing the GNAQ p.Q209L mutation. For this reason, we now include data on RNA sequencing of EC expressing GNAQ-Q209L and GNAQ-WT. MAPK signaling pathway is one of the top differentially regulated KEGG and GO-BP pathway. We generated EC lines by transducing them with lentiviral constructs promoting Doxycycline-inducible (i) expression of GNAQ-Q209L and GNAQ-WT. ERK signaling was increased only in the mutant EC GNAQ-Q209L, in a Doxycycline dose-dependent manner (see new Fig.5F).

Minor comments:

1. The scheme shown in Fig 1A and B is confusing. Please indicate for each experiment which Cre line and tamoxifen administration protocol was used.

R: We included breeding scheme and Cre-driver used in each Figure.

2. The genotype of the control mice should be specified.

R: We specified the genotype for all of the control and mutant mice with color-coded labels.

3. Some of the information is repetitive; for example, Fig 2 and 3 could be combined.

R: This manuscript has been extensively revised and these Figs are now supplemental.

4. Line 162, 163: 'Two-photon microscopy and Imaris image analysis showed the presence of perfused vascular tufts (see arrows) in the brain (Fig.1E).' Non-perfused tufts would not be visualized, but immunofluorescence analysis of brain sections after an injection with a fixable dextran could demonstrate that all lesions are perfused.

R: Thank you for this suggestion. We have corrected this in the Result section for new Suppl. Fig.S4.

Reviewer #2 (Remarks to the Author):

The authors established two mouse models in activating *Gaq11* or overexpressing *GNAQR183A* specifically in vascular endothelial cells. From the phenotype analyses, ERK1/2 activation and MEK inhibitor therapeutics, it was clearly demonstrated that *GNAQ* plays an essential role in vascular development and homeostasis.

General and major concerns: The endothelial cell-specific activation and expression were not characterized. In general, most of figures on microvessels lack high power images. Phenotypes of mice were not analyzed thoroughly, for example, the gross growth, and survival rate, and physiological changes (e.g why early lethal if it was, which was mentioned in the discussion). The work also lacks novel mechanistic insights into how *GNAQ* cause vascular malformations and solely selected ERK1/2 for study. Therefore, the study was considered as mouse model with phenotypic verifications.

R: We thank the Reviewer for this feedback. 1-We have extensively revised our manuscript and now include data on the EC-specific recombination in pups and adult mice (see Suppl. Fig. S2 and S8). 2- We provide brightfield and immunohistochemistry/immunofluorescence images for both subcutaneous and intestinal *muscularis* tissue for all of the relevant experiments (see Figs.1,2,7). 3- In this substantially revised manuscript we have characterized vascular morphogenesis defects (Figs.1-2, Suppl. Videos 1 and 2), increased EC proliferation (Fig.2, Suppl. Video 2), vascular permeability (Fig.3, Suppl. Video 3), coagulopathy (Fig.4, Suppl. Video 4), transcriptomic analysis of the *GNAQ*-mutant EC (Fig.5 Suppl. Fig.S14) and MAPK signaling in EC (Fig.5,6). Furthermore, we have extended our characterization to include Kaplan-Meier survival curves, and weight curves (see Figs.1,2,7 and Suppl. Figs.S4, S6, S18). 4- We have now included transcriptomic analysis of the *GNAQ*-mutant EC and showed that the MAPK signaling pathway is one of the top differentially regulated KEGG pathway (Fig.5). The transcriptomic analysis also highlighted upregulation of genes implicated in angiogenesis, inflammation and complement and coagulation cascades. Up to date, the mechanisms by which *Gαq* regulates aberrant vascular development and remodeling remained unknown, this manuscript provides a murine model for studying (EC-specific) *GNAQ*-related vascular tumors and related severe complications including hyperpermeability and severe coagulopathy such as KMP, these will aid in development of targeted therapies, which are highly needed to treat young patients and currently lacking.

The phenotype characterization: it was limited to a very short time frame (3-5 days after induction) without clear rationale. Any late time points were analyzed?

R: We now report Kaplan-Meier survival curves and justify the analysis time point chosen as time point corresponding to about 50% lethality.

Some specific concerns (could be applied to all figures):

Fig.1: Two photon imaging was nice. A dynamic video would help to visualize perfusion and flow within the tufts.

R: Thank you for this suggestion. Because of the extensive manuscript revisions, these data are now part of the Suppl. Data.

Fig. 2: B/C: Quantifications of vessel density in retinas was missing.

Section imaging of retinas and hyaloid would have better visualize the tufts.

Vessel diameters: were they artery, vein or microvessels? Were microphage number differed in KO groups vs WT?

R: We have tried sectioning the retinas as suggested, however it is difficult to find the exact location of the tufts and we believe the flat mount is the best approach to visualize them. To investigate the tuft location (artery versus vein-capillary), we have used the retina as model in which arteries and veins can be readily distinguished. Analysis of mutant *iEC-GNAQ^{Q209L}* mice at P8 showed that vascular tufts were exclusively localized in retinal veins and capillaries, while absent in the arteries (see new Suppl. Fig.S5). We did not analyze macrophage number as we feel it is outside the scope of this study. Hyaloid data were removed based on concerns raised by Reviewer 1 that the phenotype could reflect lack of hyaloid vessel regression rather than increased vascularity.

Some tissues were examined (from Fig.1-5). However, the study was simply vascular densities. No functional studies were performed.

R: In this revised manuscript, we included new data on functional studies to evaluate proliferation (Fig.2H-J, Suppl. Fig.S9, Suppl. Video 2), vascular permeability (Fig.3, Suppl. Video 3), coagulopathy (Fig.4, Suppl. Figs.S10-12, Suppl. Video 4). Furthermore, all these functional parameters are now evaluated in mice treated with Trametinib (Fig.7)

Fig.6 had human speciesmen were strength.

R: Thank you for the positive comment!

Fig.7 therapeutics with MEK inhibitor was nice. It would be more informative to dissect why ERK1 and if GNAQ inhibitors had better efficacy in the mouse models.

R: Thank you for the positive comment. In this revised manuscript we include transcriptomic and western blot data supporting the importance of the MAPK pathway in GNAQ mutant EC and we included analysis of the transcriptional changes driven by Trametinib. We additionally included data on a therapeutic study with the use of Trametinib on established lesions, and report extended mutant mouse survival compared to vehicle treated animals (Fig. 7N).

Reviewer #3 (Remarks to the Author):

In this manuscript, Schrenk et al. generated the first murine model for GNAQ-related vascular anomalies, exhibiting endothelial cell (EC)-specific expression of the constitutively active mutant GNAQ pQ.209L of the Gαq protein. Gαq gain-of-function mutation results in widespread abnormal vascular morphogenesis and potential thrombocytopenia and severe coagulopathy. Such vascular morphogenesis defects are consistent with findings in patients with Gαq gain-of-function mutations. The authors characterized vascular abnormalities in the brain, retina, hyaloid, subcutaneous muscle and intestinal muscle in GNAQQ209 mice at different time points of development, and confirmed MAPK/ERK activation and proliferation markers in patient-derived tissue. Finally, the authors performed proof-of-concept experiments, demonstrating that MEK inhibition with Trametinib rescued vascular and coagulopathy phenotypes.

Since mechanisms by which Gαq regulates aberrant vascular development and remodeling remain unknown, this paper provides a murine model for studying (EC-specific) GNAQ-related vascular anomalies that will aid in development of targeted therapies, which are currently lacking. The authors provide extensive evidence of aberrant vascular morphologies that recapitulate what are seen in patients. The manuscript is easy to follow, and figures containing schematics of mouse breeding and experimental timeline are helpful. However, there are important limitations in this manuscript that limit enthusiasm that are listed below as well as more minor concerns.

R: We are pleased that this Reviewer found the manuscript easy to follow and that it includes extensive evidence of vascular morphologies showing that our model recapitulates patients' phenotype. We have addressed the Reviewer's concerns below.

MAJOR CONCERNS

1. This study is very observational. There was no further insights into the functionality of the altered vasculature. How does mutant Gαq affect EC biology beyond EC proliferation? Is the endothelium more inflamed?

R: As suggested, we have extended our analysis of the mutant vasculature functionality. In this revised manuscript we have characterized 1-vascular morphogenesis defects (Fig.1-2, Suppl. Video 1), 2-proliferation (Fig.2, Suppl. Videos 2), 3-increased permeability (Fig.3, Suppl. Video 3), 4-coagulopathy (Fig.4, Suppl. Video 4) and 5-transcriptomic analysis of the GNAQ-mutant EC and EC pro-inflammatory phenotype (Fig.5). Furthermore, we have extended our characterization to include Kaplan-Meier survival curves (see Fig.1,2,7 and Suppl. Figs.S4, S6, S18).

2. There are very important controls not done in this manuscript that are needed to complete the claim that an EC-specific overexpression of GαqQ209 is responsible for the observed vascular effects. These include showing that there was no leakage of expression until tamoxifen was given. The controls in Figs 2-8 were done in mice that were littermates but did not have the GαqQ209 expression vector. It should have been done in mice with the GαqQ209 construct but without tamoxifen. Were the vessels normal until that was done? Also were the non-ECs normal and the effect was targeted to the endothelium?

Finally, what was the relative level of GαqQ209 pre and post tamoxifen versus native Gαq? Was the effect seen just because there is a massive increase in Gαq in the ECs and even normal Gαq would have resulted in the same outcome.

R: We thank the reviewer for these comments and insights.

1. As noted in response to R1 and R2, we now included no-tamoxifen mutant mice controls for most of the experiments and highlighted the specific genotype of mice used for every graph. In all of our experiments, in both pups and young adults no-tamoxifen mutant mice had normal vascular phenotype, comparable to tamoxifen treated controls or WT mice. Additionally, we have included comprehensive analysis in pups and young adult animals to test the efficiency and specificity of the *Cdh5-iCreER* driver with and without tamoxifen, by crossing *GNAQ^{Q209L}* mice with the *Rosa26^{tdTomato}* lineage reporter. There was only minimal recombination in *Cdh5-iCreERT2; tdTomato* mice that did not receive tamoxifen (1.8±0.88% in pups; 2.7±2.50% and 2.20±1.60% in adult mice subcutaneous and intestinal *muscularis* tissues; see Suppl. Figs.S2 and S8). In addition, we did not detect tdTomato expression in non-EC. No-tamoxifen mutant mice are alive at 13 months, and do not have overt phenotype or illness.

2. This is an important point. To assess that the phenotype is driven by the mutant GNAQ Q209L expression and not just GNAQ WT overexpression we have performed transcriptomic analysis of EC overexpressing GNAQ-Q209L and GNAQ-WT. The principal component analysis in Fig.5A shows that EC GNAQ-WT + Dox clusters very closely to GNAQ-WT-no-Dox, while there was a clear separation from the mutant EC GNAQ-Q209L + Dox. Furthermore, we also show in Fig. 5F that increased p-ERK is exquisitely upregulated in a Doxycycline-dependent manner in EC expressing GNAQ-Q209L, while no upregulation was noted in EC overexpressing GNAQ-WT.

3. There were two different EC-specific promoters studied. The rationale for that and why some studies were done mixing the two or with only one is not explained other than one statement that the results were not different, yet we don't know efficiency of the two in driving human GαqQ209 expression in different tissues.

R: Initial experiments in pups were performed with the *Pdgfb-iCreER^{T2}*, however due to the COVID pandemic we had to cut down the number of mouse lines and switched to the *Cdh5-iCreER^{T2}*. We have revised the manuscript and now included in the main text only data obtained with the *Cdh5-iCreER^{T2}*. In Suppl. Fig. S7 we show that vascular density, area and vessel diameter in the subcutaneous and intestinal *muscularis* tissues are comparable in *Pdgfb-iCreER^{T2}* and *Cdh5-iCreER^{T2}*-driven GNAQ mutant mice.

MINOR CONCERNS

1. The authors should discuss Huang et al. ATVB 2022. (PMID: 34670408) on EC biology and its support of the studies in this manuscript.

R: We now include this important reference.

2. A report of complete blood count and coagulation tests (PT and aPTT) in these mice as well as blood smears to look for schistocytes/polychromasia would be helpful. The effects of Trametinib on the blood counts should be compared to the non-tamoxifen-treated control discussed above as well as wildtype mice.

R: We now report complete blood count (CBC) in mutant mice (with and without tamoxifen) versus control and mutant mice treated with Trametinib (Suppl. Figs.S10, S16). Furthermore, we have analyzed PT, aPTT and blood smears (Suppl. Fig.S11), revealing reduced PT and increased polychromasia in mutant mice.

3. Page 6 – line 193: please explain the difference in subcutaneous tissue vascular area in early onset versus adult onset of GNAQ mutation. Why is it that only in early onset of GNAQ mutation that the authors observed no change in subcu. vascular area?

R: To address the reviewers' comments we have analyzed additional mutant and control mice and we now see significantly increased vascular density in both early postnatal (Fig. 1C) and adult mice (Fig. 2B-D).

4. How does mutant Gαq affect EC biology beyond cell proliferation? Is the endothelium more inflamed? Do EC express higher levels of surface adhesion molecules? What about vascular permeability? Are there any changes in blood pressure longevity?

R: We now show additional functional data that endothelial mutant Gαq expression results in increased EC proliferation (Fig. 2H-J, Suppl. Fig. S9), vascular permeability (Fig. 3, Suppl. Video 3) and coagulopathy (Fig. 4, Suppl. Video 4), while mouse survival is reduced (Fig. 1, 2, Suppl. Figs. S4, S6). We additionally analyzed EC inflammatory phenotype by RNA sequencing and qPCR and see increased expression of inflammation markers such as E-Selectin, ICAM1, IL1β, IL6 and CXCL8 (Fig.5 and Suppl. Fig.S14).

5. Please provide evidence of thrombosis in the vasculature. For example, fibrin & platelet deposition at the lesions?

R: This was also requested by Reviewer2. In Fig.4, Suppl. Fig.S12 and Suppl. Video 4 we show intralesional CD41 and CD42b platelet trapping and fibrinogen accumulation in the vascular tufts.

6. Trametinib also inhibits platelet MEK. Is the observed reverse of thrombocytopenia and coagulopathy seen in Trametinib-treated iEC-GNAQQ209L mice due to direct effect of Trametinib on platelets and other cells? Please discuss.

R: This is an important point. While we cannot exclude an effect of MEK inhibition on platelets, the transcriptomic data in Figure 5 and Suppl. Fig.S14 show an enrichment of genes associated with the Complement and Coagulation pathways as well as the Inflammatory response in the mutant GNAQ EC suggesting they provide a pro-inflammatory/pro-thrombotic environment. Upon Trametinib treatment this is normalized as shown in Suppl. Fig. S17, suggesting a main effect on the EC. We added a paragraph in the discussion.

7. Consider changing the title of the manuscript? The current title states what is already known, not really what the authors are trying to show.

R: We thank the reviewer for this suggestion to change the title of our manuscript. The title is now: MEK inhibition reduced vascular tumor growth and coagulopathy in a mouse model with endothelial hyperactive GNAQ.

8. Since both male and female mice were used, did the author observe any sex difference?

R: This is an important point. We have analyzed survival combined and separated male versus female and did not observe significant differences (see Suppl. Figs.S6 and S18).

9. An edited manuscript will be returned, but minor typos include: Use hyphen when appropriate. "Ki67-postive" not "Ki67 positive". Double check μm versus μM. Some units are mixed up. Subcutaneous "tissue", not subcutaneous "muscle". Use μL, not uL. Check spacing consistency throughout. For example, area≥ 200μm² vs. area ≥ 200μm². Mean ± SD vs. mean±SD.

R: We thank the reviewer for this note about typos. We have corrected them.

REVIEWERS' COMMENTS

Reviewer #1 (Remarks to the Author):

The manuscript provides a valuable characterization of the first genetic mouse model for GNAQ-related vascular pathology. The authors have done a great job in revising and strengthening the manuscript by adding important new experimental data and also improving the quality of imaging data.

I have only a few minor comments:

Line 206-207: "Quantification of extravasated RBC located outside of the CD31+ vascular channels revealed increased vascular permeability..": Extravasation of RBC would also indicate breakdown of vessel integrity rather than just increase in permeability?

In the last experiment (Fig. 7N), the therapeutic efficacy of Trametinib in a more advanced disease state is assessed by following the survival of the mice. It would have been of great interest to understand if Trametinib only prevents/slows down further growth of the lesions (as may be suggested by the graph), or also promotes their progression. Addition of any data on this would be valuable.

Reviewer #2 (Remarks to the Author):

The authors have addressed my concerns. In addition, they have provided much more extensive revision from the original version. The manuscript is much improved.

Reviewer #3 (Remarks to the Author):

This manuscript is a very comprehensive and convincing analysis of the role of Gαq hyperactivity in a number of vascular anomalies and the potential treatment blocking the ERK pathway. The data are strong and convincing and well-laid out. The authors appear to have been highly responsive to prior reviewers' comments.

Minor comments related to the platelet biology discussed.

1) CD42b does not detect activated platelets (lines 253-254) but detects platelets that have not undergone cleavage of the extracellular portion of GPIIb/IIIa. It does not change anything in the manuscript, but should be corrected.

2) Endothelial cells and megakaryocytes share many common gene expression. Did the treatment with tamoxifen activate the abnormal GNAQ in megakaryocytes as well as the endothelium? The suggested local infusion of tamoxifen noted in the discussion on lines 463-464 could be extended to discuss more targeted effects and eliminate this concern. It should be discussed.

Response to reviewers' comments for

Title: **MEK inhibition reduced vascular tumor growth and coagulopathy in a mouse model with endothelial hyperactive GNAQ**

Authors: Sandra Schrenk^{1,2}, Lindsay J Bischoff^{1,3}, Jillian Goines¹, Yuqi Cai¹, Shruti Vemaraju⁴, Yoshinobu Odaka^{4,5}, Samantha R Good¹, Joseph S. Palumbo^{2,6}, Sara Szabo^{2,7}, Damien Reynaud^{1,2}, Catherine D Van Raamsdonk⁸, Richard A Lang^{2,9}, Elisa Boscolo^{1,2*}

REVIEWERS' COMMENTS

Reviewer #1 (Remarks to the Author):

The manuscript provides a valuable characterization of the first genetic mouse model for GNAQ-related vascular pathology. The authors have done a great job in revising and strengthening the manuscript by adding important new experimental data and also improving the quality of imaging data.

Response: We thank this Reviewer for the good feedback!

I have only a few minor comments:

Line 206-207: "Quantification of extravasated RBC located outside of the CD31+ vascular channels revealed increased vascular permeability.": Extravasation of RBC would also indicate breakdown of vessel integrity rather than just increase in permeability?

Response: This is a good note. We edited the text accordingly.

In the last experiment (Fig. 7N), the therapeutic efficacy of Trametinib in a more advanced disease state is assessed by following the survival of the mice. It would have been of great interest to understand if Trametinib only prevents/slows down further growth of the lesions (as may be suggested by the graph), or also promotes their progression. Addition of any data on this would be valuable.

Response: This is a good question and certainly of high interest, however this would require additional experiments to be performed. During the survival experiment the collection of quality tissue for analysis was not possible as mice were found deceased or were sacrificed by the animal core veterinarians for ethical reasons.

Reviewer #2 (Remarks to the Author):

The authors have addressed my concerns. In addition, they have provided much more extensive revision from the original version. The manuscript is much improved.

Response: We thank this Reviewer for the positive comments and are delighted to know this version of the manuscript was much improved!

Reviewer #3 (Remarks to the Author):

This manuscript is a very comprehensive and convincing analysis of the role of Gαq hyperactivity in a number of vascular anomalies and the potential treatment blocking the ERK pathway. The data are strong and convincing and well-laid out. The authors appear to have been highly responsive to prior reviewers' comments.

Response: We thank this Reviewer for the great feedback.

Minor comments related to the platelet biology discussed.

1) CD42b does not detect activated platelets (lines 253-254) but detects platelets that have not undergone cleavage of the extracellular portion of GPIIb/IIIa. It does not change anything in the manuscript, but should be corrected.

Response: We thank this reviewer for pointing this out, we edited the text accordingly.

2) Endothelial cells and megakaryocytes share many common gene expression. Did the treatment with tamoxifen activate the abnormal GNAQ in megakaryocytes as well as the endothelium? The suggested local infusion of tamoxifen noted in the discussion on lines 463-464 could be extended to discuss more targeted effects and eliminate this concern. It should be discussed.

Response: A recent publication 'Kilani B, et al., 2019 Journal of Thrombosis and Haemostasis PMID: 30801958', showed that there was no recombination in adult hematopoietic stem cells including megakaryocytes in Cdh5-CreERT2 adult mice, while this was reported when using Tie2-Cre when crossed with mT/mG mice. Tamoxifen doses and regimens were similar to the ones we used in our manuscript.